# TAS-Seq is a robust and sensitive amplification method for bead-based scRNA-seq

Shigeyuki Shichino [1], Satoshi Ueha[1], Shinichi Hashimoto [2], Tatsuro Ogawa[1], Hiroyasu Aoki[1], Bin Wu[1], Chang-Yu Chen[1], Masahiro Kitabatake[3], Noriko Ouji-Sageshima[3], Noriyoshi Sawabata[4], Takeshi Kawaguchi[4], Toshitugu Okayama[5], Eiji Sugihara [6,7], Shigeto Hontsu[8], Toshihiro Ito[3], Yasunori Iwata[9], Takashi Wada[9], Kazuho Ikeo[5], Taka-Aki Sato[6] & Kouji Matsushima [1✉]

Single-cell RNA-sequencing (scRNA-seq) is valuable for analyzing cellular heterogeneity. Cell composition accuracy is critical for analyzing cell–cell interaction networks from scRNA-seq data. However, droplet- and plate-based scRNA-seq techniques have cell sampling bias that could affect the cell composition of scRNA-seq datasets. Here we developed terminator-assisted solid-phase cDNA amplification and sequencing (TAS-Seq) for scRNA-seq based on a terminator, terminal transferase, and nanowell/bead-based scRNA-seq platform. TAS-Seq showed high tolerance to variations in the terminal transferase reaction, which complicate the handling of existing terminal transferase-based scRNA-seq methods. In murine and human lung samples, TAS-Seq yielded scRNA-seq data that were highly correlated with flow-cytometric data, showing higher gene-detection sensitivity and more robust detection of important cell–cell interactions and expression of growth factors/interleukins in cell subsets than 10X Chromium v2 and Smart-seq2. Expanding TAS-Seq application will improve understanding and atlas construction of lung biology at the single-cell level.

[1] Division of Molecular Regulation of Inflammatory and Immune Diseases, Research Institute of Biomedical Sciences, Tokyo University of Science, Chiba, Japan. [2] Department of Molecular Pathophysiology, Institute of Advanced Medicine, Wakayama Medical University, Wakayama, Japan. [3] Department of Immunology, Nara Medical University, Nara, Japan. [4] Department of Thoracic and Cardio-Vascular Surgery, Nara Medical University, Nara, Japan. [5] National Institute of Genetics, Shizuoka, Japan. [6] Research and Development Center for Precision Medicine, University of Tsukuba, Ibaragi, Japan. [7] Center for Joint Research Facilities Support, Research Promotion and Support Headquarters, Fujita Health University, Aichi, Japan. [8] Department of Respiratory Medicine, Nara Medical University, Nara, Japan. [9] Division of Infection Control, Kanazawa University Hospital, Department of Nephrology and Laboratory Medicine, Kanazawa University, Ishikawa, Japan. ✉email: koujim@rs.tus.ac.jp

Single-cell RNA sequencing (scRNA-seq) has been deciphering cellular subsets in various species, organs, and conditions in an unsupervised manner and drives the construction of single-cell atlas, such as Human Cell Atlas[1]. The primary output of scRNA-seq data in an analyzed sample is the gene-expression pattern of single cells, their classification by gene-expression similarity, and their cellular composition. Cellular composition, i.e., the abundance of transcriptionally distinct cell subsets, is a factor reflecting the functions of the analyzed sample; hence, the accuracy of scRNA-seq data cellular composition is essential to elucidate biological issues and build single-cell atlas using scRNA-seq datasets.

10X Genomics Chromium, a microdroplet-based high-throughput scRNA-seq platform, is widely used because it is user-friendly and commercially available[2]. Another major microplate/cell sorter-based scRNA-seq platform is Smart-seq2[3], often combined with a microdroplet-based system to achieve more gene-level sensitivity for every single cell[4]. However, both techniques have cell sampling bias that could affect the cell composition of scRNA-seq datasets. For example, human neutrophils dropout frequently occurs in 10X Chromium system[5], and fragile cells, such as macrophages and some types of stromal cells, tend to be lost during cell sorting because of high-pressure[5,6]. In addition, most high-throughput scRNA-seq methods use template-switching reaction[2] for cDNA amplification. Thus, efficiency is affected by the 5' structure of RNA[7], limiting the capability of analyzable RNA specimens in scRNA-seq analysis.

Terminal transferase (TdT) is a template-independent polymerase that could efficiently add homopolymer tails against 3' ends of DNA. TdT-based scRNA-seq methods rely on the homopolymer tailing reaction. Tang et al. analyzed the transcriptome of a single cell by high-throughput sequencing for the first time in 2009[8]. They used polyA tailing of the 3' cDNA end by TdT, then synthesized the second strand by oligo-dT with a tagging primer. However, because of the high processivity of the TdT enzyme[9], the Tang method produces long by-products derived from remaining free primer, which interferes with cDNA amplification and downstream analysis. To overcome this problem, Sasagawa et al. developed Quartz-seq/Quart-seq2, which minimizes by-product synthesis by the combination of exonuclease I treatment (Quartz-seq) or column purification (Quartz-seq2), along with strict control of the TdT reaction time. Huang et al. and Matsunaga et al. also developed bead sequencing dependent on TdT-based cDNA amplification with oligo-dT immobilized magnetic beads, which minimize by-product synthesis by controlling primer density on the magnetic beads through overall bead quantity in the reaction[10,11]. Because TdT reaction efficiency is higher than that of the template-switching method[7,12], Quartz-seq2 is much more sensitive than commonly used scRNA-seq methods[13]. However, stringent control of the TdT reaction, including controlling reaction time on the second scale and/or primer density on the cDNA-immobilized magnetic beads, is still necessary to avoid excessive primer-derived biproduct synthesis and cDNA amplification failure[10,14,15]. This complicates the handling TdT-based scRNA-seq methods, limiting their utility. To overcome these problems, we developed terminator-assisted solid-phase cDNA amplification and sequencing, termed TAS-Seq, a TdT-based cDNA amplification method for nanowell/bead-based scRNA-seq methods (Fig. 1). By using dideoxycytidine-mediated stochastic chain termination reaction, TAS-Seq showed high tolerance to variations in the TdT reaction. In murine and human lung samples, TAS-Seq yielded scRNA-seq data that were highly correlated with flow-cytometric data, showing higher gene-detection sensitivity and more robust detection of important cell–cell interactions and growth factors/interleukins than other widely used methods, such as 10X

Chromium v2 (10X v2) and Smart-seq2. Because it showed efficient performance with the use of only simple materials and equipment, TAS-Seq might be highly applicable to build precise single-cell atlas and associated cell–cell interaction networks.

## Results

**Dideoxycytidine spike-in during TdT-tailing reaction effectively suppresses elongation of primer-derived products in various reaction conditions.** We developed TAS-Seq for nanowell/bead-based scRNA-seq methods because they have flexibility of the reaction/buffer conditions, could isolate single cells gently by gravity flow[2], and possibly capture cell composition more precisely. We used BD Rhapsody as a nanowell/bead-based scRNA-seq system because of its commercial availability.

TdT accepts dideoxynucleotide triphosphates (ddNTPs) as substrate. When ddNTPs are incorporated by TdT, elongation of the 3′ terminus of DNA by TdT is stopped because dideoxynucleotides lack a 3′ hydroxyl group[16]. We speculated that spiking ddNTP into the homopolymer tailing reaction by TdT would terminate the tailing reaction in a stochastic manner, independently of the reaction time and TdT amount. To assess whether the ddNTP spike-in approach could increase the robustness of the TdT-based cDNA amplification, we performed a TdT-tailing reaction using deoxycytidine triphosphate (dCTP) with a 1/20 dideoxycytidine triphosphate (ddCTP) spike-in. We used potassium cacodylate buffer supplemented with $Co^{2+}$ ion, previously reported as the most efficient buffer system of the TdT-tailing reaction[12,17]. We first compared the extension length of undigested reverse transcription primers by TdT reaction between the ddCTP spike-in and dCTP alone reactions by applying exonuclease I-treated magnetic beads of BD Rhapsody. We found that ddCTP:dCTP (1:20) effectively suppressed undigested primer-derived products extension (under 200 bp) under different TdT reaction times (5 or 30 min) and quantities (10, 15, and 42 U/μL) (Fig. 2a). However, the extension of undigested primer-derived products exceeded 200 bp and reached more than 600 bp when ddCTP was not added (Fig. 2a), suggested that our TAS-Seq approach is applicable to a wider range of conditions than the dCTP only reaction.

Next, to compare the tolerance of TdT reaction in terms of the extension of remaining reverse transcription primers with the other TdT-based cDNA amplification methods, we performed a Quartz-seq2 reaction without any RNA (only reverse transcription primers). We found that the length of remaining primer-derived products was less than 200 bp under standard Quartz-seq2 condition, similar to that determined using TAS-Seq (Fig. 2b). When the amount of TdT was increased 1.5-fold, the amount of primer-derived by-products also increased, but the size distribution was not different from that under standard conditions (Fig. 2b). Furthermore, when the TdT reaction was pre-incubated at 23 °C for 5, 10, or 15 min (simulating the case of accidental reaction warming), the length of primer-derived by-products was greater than 200 bp and correlated with incubation time (Fig. 2b), suggesting that our TAS-Seq approach is more tolerable of varying TdT conditions than Quartz-seq2 in terms of the suppression of excessive extension and production of primer-derived products. In addition, on cDNA-immobilized, Exonuclease I-treated BD Rhapsody beads, ddCTP addition also effectively suppressed undigested primer-derived products extension at ranges up to 45 min of TdT reaction with visible cDNA products (Fig. 2c). Using 6000 single cells of the murine lung, TAS-Seq yielded over 2 μg of amplified cDNA with typical size distribution (peaked around 1kbp) by 16 cycles of PCR (Fig. 2d). These results indicated that TAS-Seq could amplify cDNA effectively with well-tolerated TdT reaction time and TdT

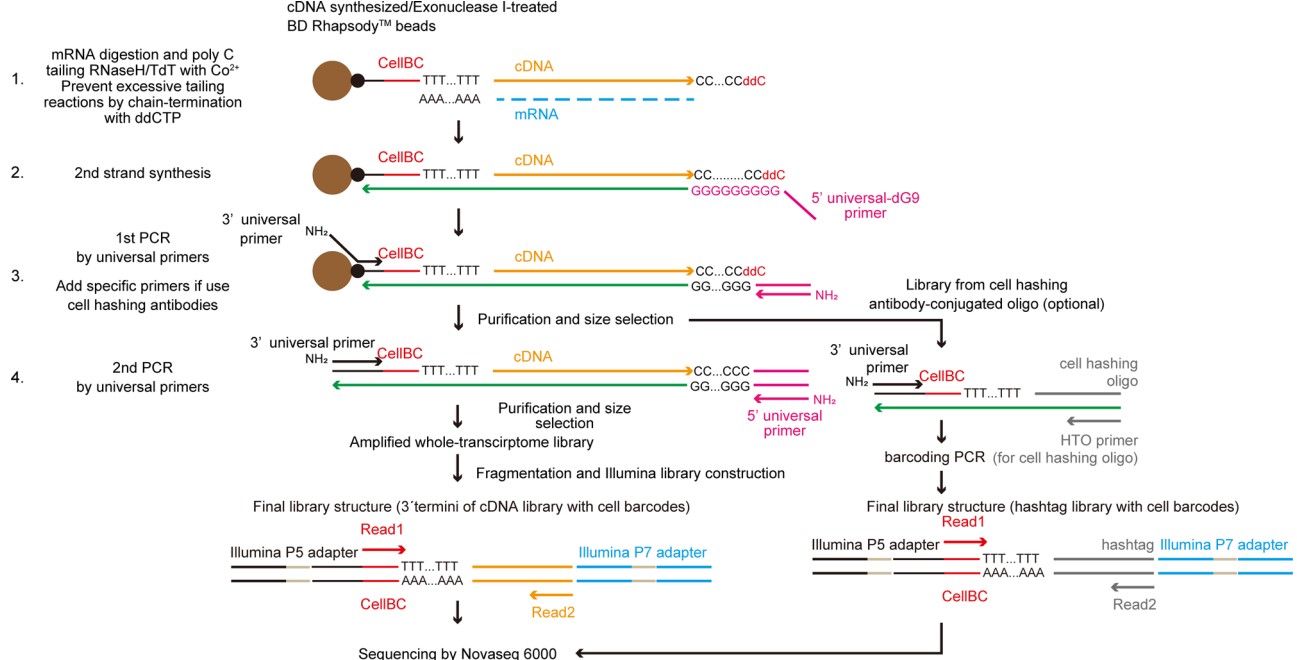

**Fig. 1 Principles of TAS-Seq.** Diagram of the TAS-Seq library preparation workflow. First, cDNA synthesis on BD Rhapsody magenic beads were performed. After Exonuclease I treatment, dC-tailing reaction was performed by TdT/RNase H. Tailing reaction was stochastically stopped by ddCTP incorporation into 3' termini (1). Next, second-strand synthesis was performed using 5'universal-dG9 primer and PCR master mix (2). Then, 1st PCR was performed by appropriate primers (3' and 5' universal primer (if only amplify cDNA) or 3', 5' universal primer and HTO primer (if use cell hashing antibodies)) (3). Resultant libraries were size-selected and amplified by 2nd PCR (4). cDNA libraries were fragmented, ligated, and truncated with Illumina P7 adapters, and 3' termini of cDNA molecules with cell barcodes were enriched by PCR amplification. During the PCR step, we introduced unique-dual barcodes to the Illumina adapters to minimize index hopping. Finally, libraries were sequenced by Illumina Novaseq 6000.

activity, which might be affected by the lot-to-lot variability of the TdT enzyme.

**TAS-Seq is compatible with cell hashing technology.** Because cell hashing by short oligonucleotide-conjugated antibodies is widely used to reduce scRNA-seq cost, we further examined whether TAS-Seq was compatible with the cell hashing method. We pooled 14 samples of BioLegend Hashtag-A labeled CD45.2[+] cells from a murine subcutaneous tumor model of Lewis lung carcinoma and subjected them to TAS-Seq as previously described[18]. TAS-Seq successfully obtained cDNA and hashtag libraries (Supplementary Fig. 1a). The demultiplexing of 14 libraries by hashtag readout revealed that cell number and genes were detected similarly among 14 samples (Supplementary Fig. 1b). Notably, library size of the Hashtag products without sequencing adapters is 175 and 132 bp for TAS-Seq and Quartz-seq2, respectively. Thus, the size distribution of primer-derived by-products of Quartz-seq2 overlapped with that of the Hashtag products, whereas that of TAS-Seq overlapped only minimally (Fig. 2a, b). These results indicated that TAS-Seq exhibited more robust compatibility with cell hashing than Quartz-seq2.

**TAS-Seq detects more genes and highly variable genes than 10X v2 and Smart-seq2 in murine lung, spleen, and kidney tissues.** To evaluate the performance of TAS-Seq, we first compared TAS-Seq with a commercial whole-transcriptome amplification (WTA) BD Rhapsody kit, a random priming-based cDNA amplification, using mouse spleen cells (Supplementary Fig. 2a). We detected significantly more genes using TAS-Seq (median 2200.8 ± 28.3 genes with median 25418.5 ± 38.2 reads/cell) than using the BD WTA kit (median 1714.2 ± 83.2 genes with median

25556.7 ± 116.8 reads/cell) (Supplementary Fig. 2b). To assess the library quality of the datasets, we analyzed the proportion of mitochondrial and ribosomal RNA gene read counts in total read counts. Both TAS-Seq and BD WTA kit datasets contained a very low quantity of ribosomal RNA gene read counts. In contrast, TAS-Seq data showed a significantly lower proportion of mitochondrial genes than BD WTA data (Supplementary Fig. 2c). We also compared the number of highly variable genes (identified by FindVariableFeatures function [selection.method = mvp] in Seurat v4.0.3[19] package), and found that TAS-Seq identified more highly variable genes than the BD WTA kit (Supplementary Fig. 2d). These results showed that TAS-Seq could detect more overall and highly variable genes and fewer mitochondrial genes than the random priming-based approach in adult murine spleen cells.

Next, we compared scRNA-seq data of single-cell suspension of adult murine lungs obtained by TAS-Seq with publicly available Smart-seq2/10X v2 data from Tabula Muris Consortium (deep-sequenced datasets)[4] and Raredon et al. (a shallow-sequenced dataset, GSM3926450)[20,21]. To evaluate technical variations of TAS-Seq, we split BD Rhapsody beads into three groups after reverse transcription and exonuclease I digestion, and amplified cDNA independently (Fig. 3a). We compared shallow-sequenced data by downsampling raw fastq data of each dataset (~23,000 mean reads/cell). In bulk RNA-seq, the methods counting the 3'-ends of RNA detected a similar number of genes to that identified using full-length RNA-seq at range from 1/10 to 1/3 sequencing depth[22]. Therefore, to compare deep-sequenced data, we set a sequencing depth of 10X v2 and TAS-Seq depth to ~1/5 of the Smart-seq2 depth. We found that TAS-Seq datasets detected more genes and Seurat-defined highly variable genes than the other datasets, both under deep and shallow sequencing conditions with minimal technical variations (Fig. 3b–e). When

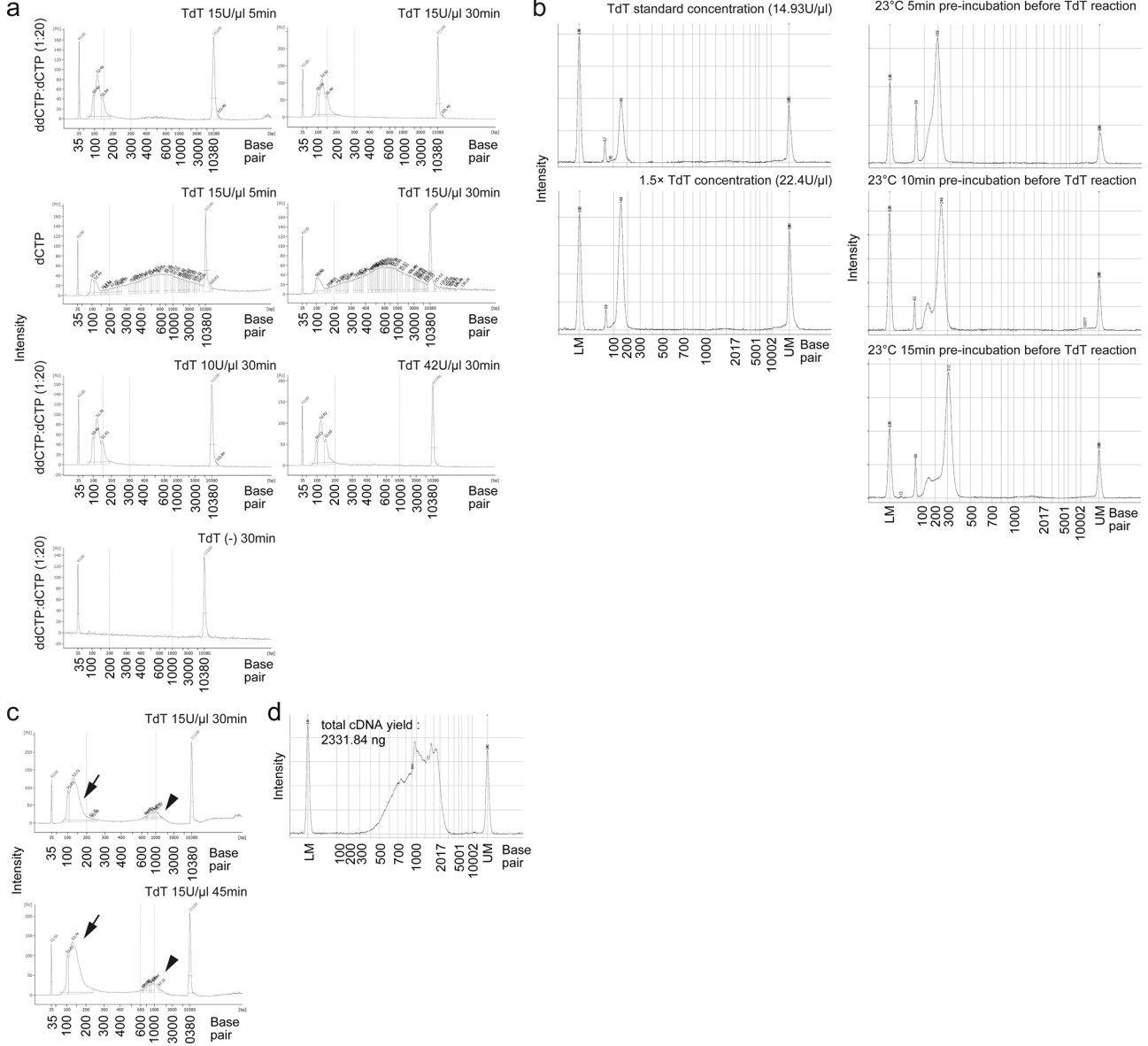

**Fig. 2 TAS-Seq effectively suppresses excessive elongation of primer-derived by-products under various TdT reaction conditions. a** and **c** TAS-Seq tolerance against TdT reaction time and TdT activity. TdT reaction with ddCTP:dCTP (1:20) or dCTP only with $Co^{2+}$ supplementation was performed from 5 or 30 or 45 min with different TdT enzyme amounts against exonuclease I-treated BD Rhapsody beads (**a**) or cDNA-synthesized, exonuclease I-treated BD Rhapsody beads (**c**). Size distributions of the 1st PCR products were analyzed. Note that the length of primer-derived bi-products (arrows) peaked at around 136 bp and did not extend over 200 bp in every reaction time. In addition, amplified cDNA was also visible (**c**, arrowheads). **b** Quartz-seq2 tolerance against TdT reaction warming and enzyme activity. Size distribution of remained primer-derived by-products of Quartz-seq2 under standard conditions, and under pre-incubation at 23 °C for 5, 10 min, and 15; 1.5× TdT amount shown. Reactions were performed without RNA. Note that primer-derived by-products were extended over 200 bp with 23 °C pre-incubation. **d** Size distribution of TAS-Seq amplified cDNA library of 6000 single cells derived from the murine lung. **a–d** Representative results of two independent experiments are shown.

compared to Smart-seq2 data, which is based on full-length RNA-seq, TAS-Seq detected more overall and highly variable genes in ~1/5 sequencing depth (Fig. 3b, c). We further compared scRNA-seq data of single-cell suspension of adult murine kidney and spleen obtained by TAS-Seq with Smart-seq2/10X v2 data from Tabula Muris Consortium[4]. Similar to murine lungs, TAS-Seq detected more genes and highly variable genes than the other datasets in mouse whole kidney (Supplementary Fig. 3a, b) and spleen (Supplementary Fig. 3c–e). These results indicated that TAS-Seq might detect more genes and highly variable genes than 10X v2 and Smart-seq2, at least in adult murine lung, kidney, and spleen in steady-state conditions.

**TAS-Seq precisely detects cell composition in murine and human lung tissue**. We next evaluated TAS-Seq performance in terms of cell type identification and quantification in murine lung tissue. Cell types of the cell clusters which were annotated manually by their marker gene expression patterns (Supplementary Data 1), and visualized by Fast Fourier transform-accelerated interpolation-based t-stochastic neighbor embedding (FIt-SNE) (Fig. 4a). We found that T cell subpopulation ($CD4^+$ and $CD8^+$ T cells) showed better separation in TAS-Seq data than in the other datasets, and deep-sequencing data showed better separation than shallow sequencing data within TAS-Seq datasets (Fig. 4a, arrowheads), suggesting that TAS-Seq achieved

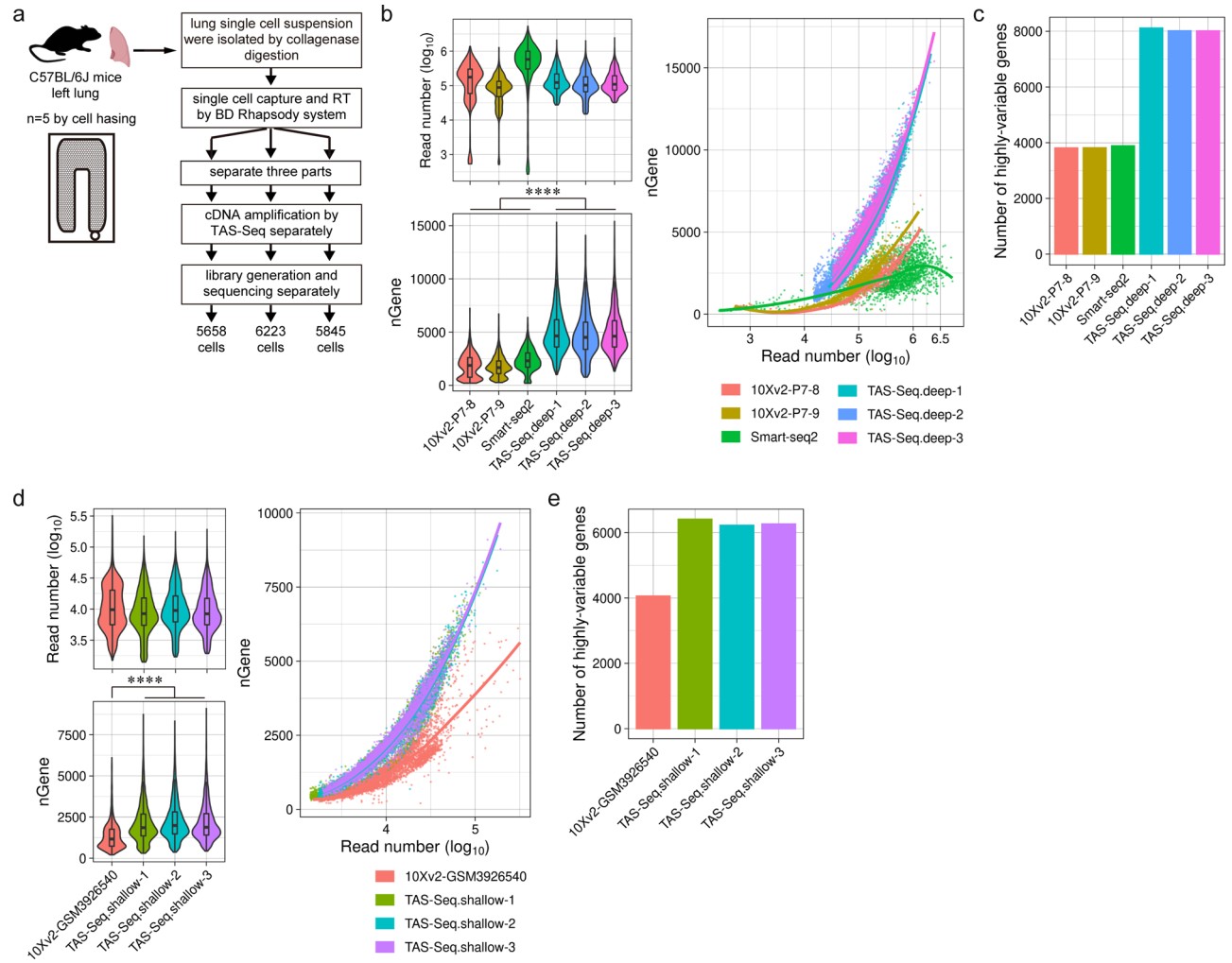

**Fig. 3 TAS-Seq detects more overall and highly variable genes than 10X v2 and Smart-seq2 in murine lungs with minimal technical variability.**
**a** Experimental scheme of TAS-Seq library generation from the lungs of adult C57BL/6 J mice. A lung single-cell suspension was obtained and processed using the BD Rhapsody workflow until exonuclease I treatment. Resulting beads were then split into three groups and TAS-Seq was performed individually. **b** and **d** Violin/box plot of the read number and detected gene number; scatter plot of the read number/detected gene number for each cell in TAS-Seq (deep-sequenced), 10X v2 (Tabula Muris), and Smart-seq2 datasets (**b**) or TAS-Seq (shallow-sequenced) and 10X v2 (GSM3926540) datasets (**d**). Box plot shows the mean of the read number with upper and lower quantiles, and the whisker shows ±1.5 × interquartile range. ****$p = 0$ (two-sided Wilcoxon rank-sum test). **c** and **e** The number of highly variable genes in TAS-Seq (deep-sequenced), 10X v2 (Tabula Muris), and Smart-seq2 datasets (**c**) or TAS-Seq (shallow-sequenced) and 10X v2 (GSM3926540) datasets (**e**). Exact $p$-values and statistics are shown in Supplementary Data 5.

better separation of murine lung T cell subsets than Smart-seq2 and 10X v2 platforms.

To evaluate the accuracy of cell composition quantification in adult murine lung tissue by each scRNA-seq platform, we compared cell composition data obtained by scRNA-seq and flow-cytometric analysis (gating schemes for flow-cytometric data are shown in Supplementary Fig. 4a–c). TAS-Seq results showed minimal differences between three technical replicates under both shallow- and deep-sequencing conditions (Fig. 4b, d). Next, we evaluated the cell composition between different biological replicates of TAS-Seq data. We found that the difference of cell composition between two biological replicate data (TAS-Seq.dataset2 and the other TAS-Seq datasets) was higher than among technical replicate data, but the tendency of cell composition was still similar (Fig. 4b, d). Surprisingly, cell compositions determined by different scRNA-seq platforms varied (Fig. 4b–d). TAS-Seq data showed the highest Pearson's correlation coefficient ($R^2 = 0.979 \pm 0.00165$ in shallow-sequenced data, $R^2 = 0.973 \pm 0.00442$ in deep-sequenced data, and $R^2 = 0.912$ in dataset2), followed by Smart-seq2 ($R^2 = 0.886$), 10X v2 P7-8 ($R^2 = 0.649$), 10X v2 GSM3926450 ($R^2 = 0.563$), and 10X v2

P7-9 ($R^2 = 0.548$) (Fig. 4c). TAS-Seq data also showed the slope of the regression line was nearest to 1 among the compared platforms (Supplementary Fig. 5). We also found that 10X v2 datasets under-represented fibroblast fractions and over-represented alveolar macrophages, whereas Smart-seq2 data over-represented endothelial cells and monocytes than flow-cytometric data (Fig. 4d). Notably, alveolar macrophages were lost in Smart-seq2 data (Fig. 4d). These results indicated that TAS-Seq might acquire more accurate cellular composition data than Smart-seq2 and 10X v2 in steady-state adult murine lungs.

We further analyzed human lung samples of fibrotic and non-fibrotic tissues from a rheumatoid arthritis-associated interstitial lung disease (RA-ILD) patient by TAS-Seq. Cell clustering analysis revealed that TAS-Seq captured the difference of cell composition between fibrotic and non-fibrotic areas from the same patient with minimal batch-effect (Fig. 4e). Similar to results from murine lung tissue, TAS-Seq obtained scRNA-seq data highly correlated with flow-cytometric data ($R^2 = 0.937$ in fibrotic area; $R^2 = 0.942$ in non-fibrotic area) in RA-ILD samples (Fig. 4f, g, gating schemes are shown in

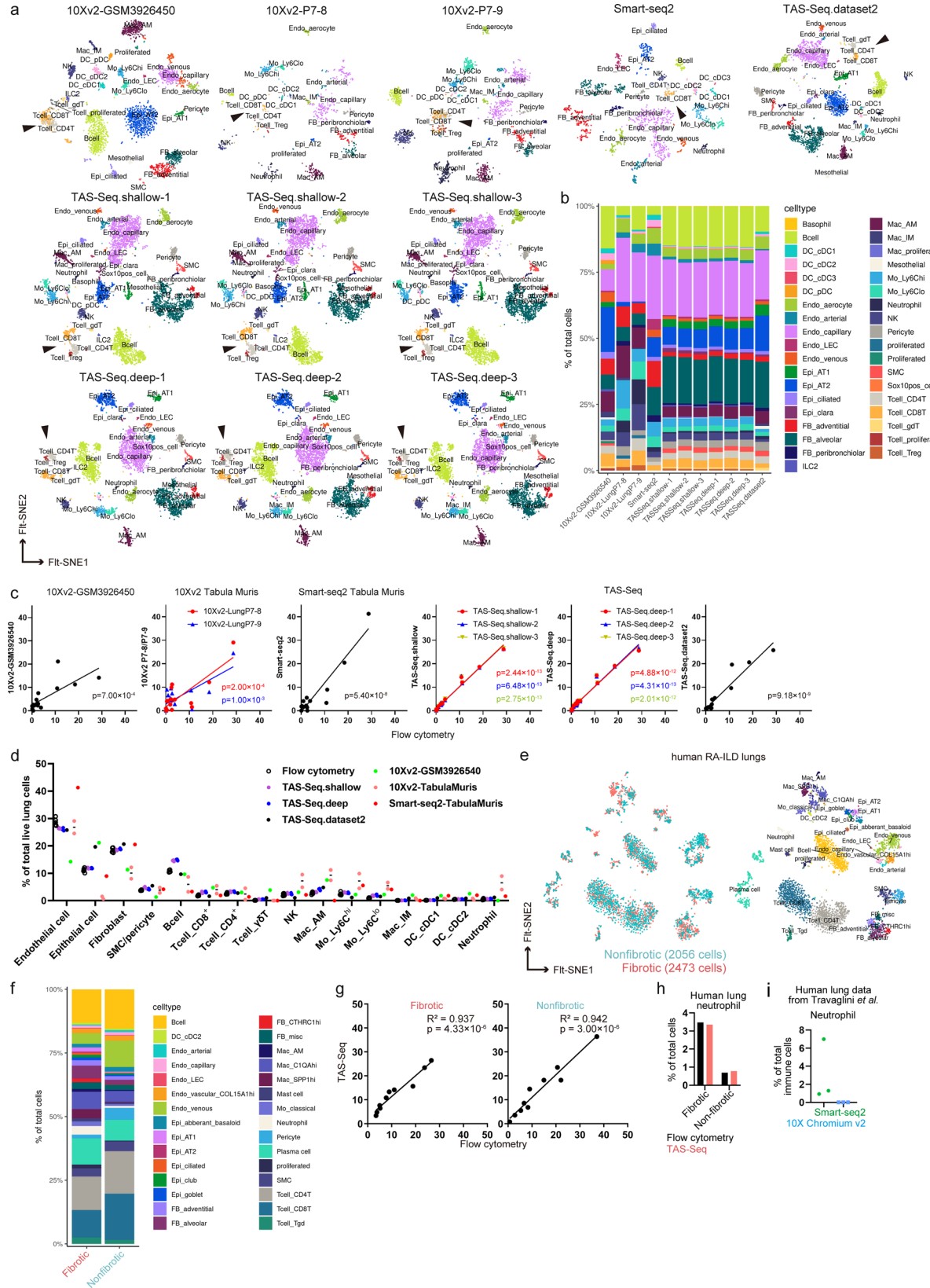

Supplementary Fig. 6a–b). Strikingly, TAS-Seq precisely detected neutrophils depleted in the 10X v2 dataset of human lungs[5] (Fig. 4h–i). These data indicated that TAS-Seq could capture cell composition of adult murine lungs and human RA-ILD lungs more precisely than Smart-seq2 and 10X v2, with high gene-detection sensitivity.

**TAS-Seq detects more genes with a lower drop-out rate than Smart-seq2 and 10X v2 in murine lung cell subsets.** Because the gene-detection rate was different between different lung cell subsets, we next compared the number of detected genes in TAS-Seq, Smart-seq2, and 10X v2 datasets within commonly detected cell subsets. TAS-Seq detected significantly more genes than the

**Fig. 4 TAS-Seq accurately detects cell composition of murine and human lungs. a** Visualization of cell clustering of each scRNA-seq dataset of the murine lung by Seurat v4.0.3 package in 2D Flt-SNE space. Note that CD4$^+$ and CD8$^+$ T cells were more clearly separated in Flt-SNE space in both shallow- and deep-sequenced TAS-Seq data than in the other datasets. **b** Stacking plot shows the composition of each annotated cell in each scRNA-seq dataset from murine lung tissue. **c** and **d** Comparison of cell composition between flow-cytometric data and scRNA-seq datasets of the murine lung. *p*-values of Pearson's correlation coefficients are shown. Pearson's correlation coefficients and slope of regression lines are shown in Supplementary Fig. 5. **e** Cell clustering in each scRNA-seq dataset from human RA-ILD lungs by Seurat v4.0.3 package in 2D Flt-SNE space. Note that minimal batch effects were seen between fibrotic- and non-fibrotic lung samples. **f** Stacking plot shows the composition of each annotated cell per scRNA-seq dataset from the human RA-ILD lung. **g** Comparison of cell composition between flow-cytometric data and scRNA-seq datasets of the human RA-ILD lung. *p*-values and $R^2$-values of Pearson's correlation coefficients are shown. **h** The composition of neutrophils of flow-cytometric and TAS-Seq data in RA-ILD lungs. **i** Composition of neutrophils in Smart-seq2 and 10X v2 datasets of human lungs reported by Travaglini et al. Details of the cell annotations and associated marker genes of each mouse and human dataset are shown in Supplementary Data 1.

other datasets within the commonly detected cell subsets at a given sequencing depth (Supplementary Fig. 7a, b). We further evaluated the drop-out rate of detected genes in the lung cell subsets and found that rates were lower in TAS-Seq data than in 10X v2 or Smart-seq2 data (Supplementary Fig. 8a, b). In addition, genes detected only in TAS-Seq data were expressed at lower levels than the other genes (Supplementary Fig. 8a, b), suggesting that TAS-Seq better captured low-expression genes than 10X v2 and Smart-seq2 in cell subsets of steady-state adult murine lung tissue. To investigate possible bias in TAS-Seq data, we analyzed the percentage of GC content and transcript length of detected genes in each murine lung cell subset. We found that TAS-Seq favorable genes had less GC content than the commonly detected genes identified with 10X v2 or Smart-seq2 (Supplementary Fig. 9a, b), and had longer transcript length than the commonly detected genes detected by 10X v2 (Supplementary Fig. 10a, b), suggesting that TAS-Seq favorably detects long and/or AT-rich genes in murine lung cell subsets.

Next, we analyzed characteristics of read count distribution for each detected gene in TAS-Seq, 10X v2, and Smart-seq2 datasets. In pseudo-bulk count data of each cell subset, we found that major parts of detected genes were highly expressed in 10X v2 and Smart-seq2 data (Supplementary Fig. 11a). In contrast, the read count of detected genes in TAS-Seq data were distributed more uniformly against gene-expression levels than that of 10X v2 and Smart-seq2 data among analyzed cell subsets, and genes with low expression were detected more frequently in deep-sequenced TAS-Seq data (Supplementary Fig. 11a). We further compared the distribution of lung fibroblasts with that seen in bulk RNA-seq data of sorted murine lung fibroblasts using our previously published data (GSE110540). Similarity calculation of the density distribution via Kullback-Leibler divergence revealed that the similarity between TAS-Seq and bulk RNA-seq data was higher than that between 10X v2 or Smart-seq2 and bulk RNA-seq data (Supplementary Fig. 11b, c). These results indicated that TAS-Seq detects genes with varying expression more uniformly than 10X v2 and Smart-seq2.

**TAS-Seq detects important cell–cell communications in adult murine lungs.** Cell–cell interaction network analysis is a major downstream analysis of scRNA-seq data and is possibly affected by the cell composition accuracy and drop-out rate of expressed genes in each cell subset of scRNA-seq datasets. Using CellChat software[23] that considers the abundance of cell subsets, we inferred cell–cell interactions of adult murine lungs using TAS-Seq (shallow and deep-sequenced datasets), Smart-seq2, 10X v2 (shallow dataset: GSM3926540, deep dataset: pooled Tabula Muris datasets), of which total cell number was downsampled to 1717 cells (cell number of the Smart-seq2 dataset). We found that the number of inferred interactions and pathways was highest in deep-sequenced TAS-Seq data and second highest in shallow-sequenced TAS-Seq data, from soft to hard thresholds of ligand/

receptor genes (minimum percent of expressed cells in each cell subset were from 5% to 75%) within cell subsets (Fig. 5a). Of note, some of the important pathways for lung development, homeostasis, and repair, including sonic hedgehog, WNTs, bone morphologic proteins (BMPs), fibroblast growth factor (FGFs), transforming growth factor (TGF), colony-stimulating factor (CSF), and Notch signaling[24], were lost in 10X or Smart-seq2 datasets when the expression threshold became stricter (Supplementary Fig. 12a). Furthermore, these pathways were better retained in deep-sequenced than in shallow-sequenced TAS-Seq datasets, suggesting that TAS-Seq could detect important cell–cell interaction pathways more robustly than 10X v2 and Smart-seq2 when combined with CellChat analysis. In addition, CellChat analysis revealed that outgoing and incoming signaling strength of each cell subsets was higher in TAS-Seq datasets than the other datasets. Moreover alveolar type 2 epithelial cells (AT2 cells), capillary endothelial cells, and alveolar fibroblasts were the major producers/receivers within inferred cell–cell interaction network from TAS-Seq and Smart-seq2 datasets, but not in 10X v2 datasets (Fig. 5b). Moreover, AT2 cells, alveolar fibroblasts, and vascular endothelial cells were connected stronger within the CellChat-predicted cell–cell interaction network of the TAS-Seq dataset than the other datasets (Fig. 5c and Supplementary Fig. 12b). Because AT2 cell-alveolar fibroblast interaction is thought to be crucial for alveolar homeostasis, repair, and regeneration[25], TAS-Seq may detect important intercellular communication of murine lung more robustly than Smart-seq2 and 10X v2.

**TAS-Seq robustly detects growth factor and interleukin expression in adult murine lung cell subsets.** Identification of specific cell subsets that significantly contribute to specific gene expression is another important output from unsorted scRNA-seq datasets. To clarify the contribution of each cell subset to each gene expression throughout the mouse lung, we investigated the gene-expression patterns of growth factors and interleukins in each cell subset using TAS-Seq, 10X v2, and Smart-seq2 datasets of steady-state mouse lungs. We found that TAS-Seq more broadly detected high-to-low levels of growth factors among lung cell subsets, including BMPs, Csf1, FGFs, PDGFs, TGF, and VEGFs, than the other sequencing methods (Fig. 6a). All of the datasets demonstrated that *Pdgfa* and *Pdgfb*, important mediators of lung regeneration and fibrosis[26], were mainly expressed by pericytes/epithelial cells, and aerocytes/ capillary endothelial cells, respectively, but the percentage of cells expressing *Pdgfb* was highest in TAS-Seq data (Fig. 6a). In addition, all datasets showed that *Fgf7* and *Fgf10*, which are important factors for alveologenesis[27,28], were highly expressed in alveolar, adventitial, and peribronchiolar fibroblasts, and the percentage of expressed cells was also highest in the TAS-Seq data (Fig. 6a). When focusing on interleukins, TAS-Seq and 10X v2, but not Smart-seq2 data, showed broad expression of *Il2*, an

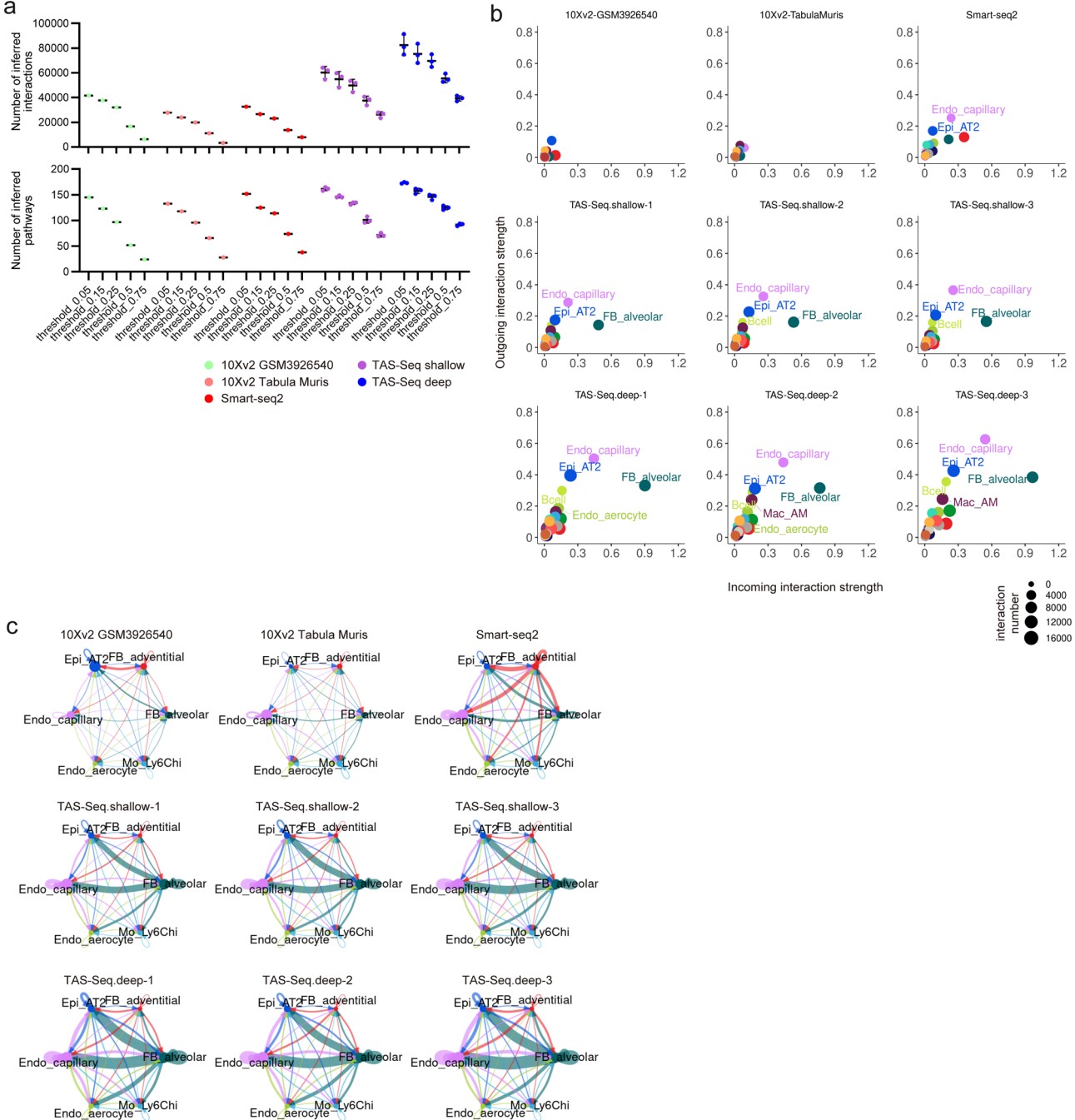

**Fig. 5 TAS-Seq identifies more cell–cell interactions than 10X v2 and Smart-seq2 in murine lung tissue. a** Changes of the number of inferred interactions/pathways of cell–cell interaction network of each scRNA-seq dataset of murine lung predicted by CellChat when the threshold of genes of which minimum fraction of expressed cells within each cell subset. Error bars show mean ± SD in TAS-Seq.shallow and TAS-Seq.deep samples. **b** Scatter plot of incoming (target) and outgoing (source) signaling strength within the cell–cell interaction network of each cell subset (minimum expression of genes in each cell subset ≥ 0.15). Dot size represents the sum of incoming and outgoing signals in each cell subset. Capillary endothelial cells, alveolar type 2 cells, and alveolar fibroblasts were strongly connected in the TAS-Seq and Smart-seq2 dataset networks, and cell subsets were more strongly connected in the TAS-Seq dataset than in the Smart-seq2 dataset. The overall connections were weak in 10X v2 datasets. **c** Circle plot visualizations of cell–cell interaction network of each scRNA-seq dataset of murine lung within particular cell subsets. Circle sizes represent the cell number of each subset. Wider edge means stronger communication, and edge width was normalized among all datasets. See also Supplementary Fig. 12b for the cell–cell interaction network of all cell subsets. Abbreviations of cell subsets were shown in Supplementary Data 6.

important cytokine for T cell activation[29], in lung cell subsets (Fig. 6b). In addition, TAS-Seq data showed ILC2/basophils and interstitial macrophages as a major source of pro-inflammatory cytokine *Il13* and anti-inflammatory cytokine *Il10* expression, respectively (Fig. 6b), consistent with previous findings[30–32]. Moreover, only TAS-Seq revealed broad expression of *Il11*, an

important cytokine in lung injury and fibrosis[33], in various lung cell subsets, consistent with a previous report[34]. These data suggest that TAS-Seq captures expression patterns of growth factors/interleukins and their major cellular sources more robustly than 10X v2 and Smart-seq2 in steady-state murine lung tissue.

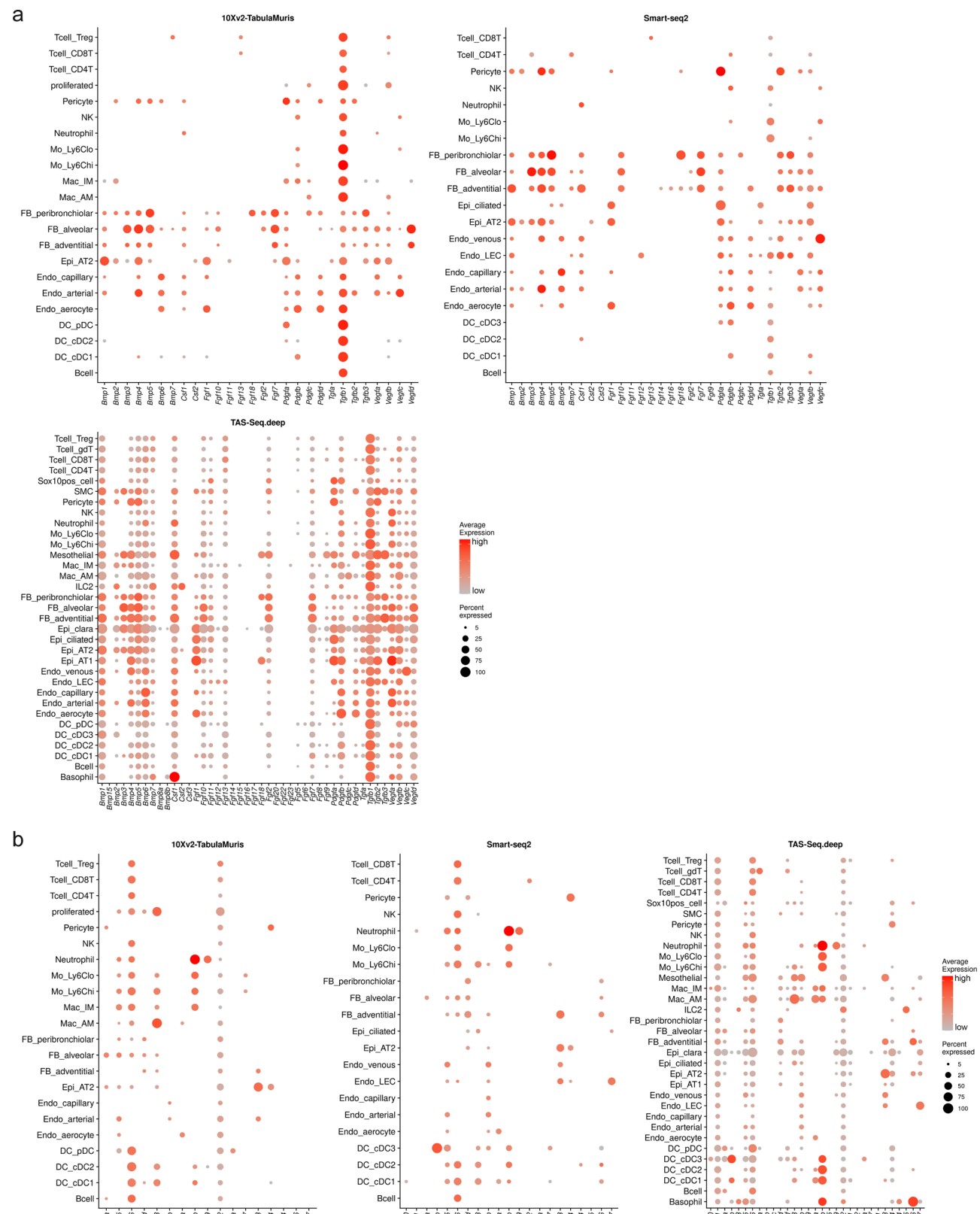

**Fig. 6 Expression patterns of growth factors and interleukins in TAS-Seq, 10X v2, and Smart-seq2 data from each cell subset of murine lung tissue.**
**a**, **b** Dot plots show expression patterns of selected growth factors—fibroblast growth factors (FGFs), bone morphogenic proteins (BMPs), platelet-derived growth factors (PDGFs), transforming growth factors (TGFs), vascular endothelial growth factors (VEGFs), and colony-stimulating factors (CSFs) (**a**)—and interleukins (**b**) in each lung cell subset in TAS-Seq (deep-sequenced), 10X v2 (Tabula Muris), and Smart-seq2 datasets. Colors represent expression level, and size of the dots represents the percent of expressed cells.

## Discussion

In this study, we developed a TdT-based scRNA-seq method called TAS-Seq. The core principle of this technique was stochastic termination of a TdT-tailing reaction by dideoxycytidine spike-in, which increased robustness of the TdT reaction by limiting excess elongation of primer-derived by-products. In addition, we found that TAS-Seq detected more overall and highly variable genes than 10X v2 and Smart-seq2 in mouse lung, spleen, and kidney tissue. TAS-Seq also yielded scRNA-seq data that were highly correlated with flow-cytometric data, showing a lower drop-out rate and more robust detection of important cell–cell interactions and growth factors/interleukin expression patterns in lung cell subsets than 10X v2 and Smart-seq2.

We showed that the TAS-Seq approach, which stochastically terminated the polyN tailing reaction of TdT, did not alter the extension of primer-derived by-products within the reaction time range (5–45 min), nor under varying TdT enzyme quantity (up to 4.2-fold increase). In addition, we showed that the length of primer-derived by-products of TAS-Seq was shorter than those of cell hashing antibodies, whereas Quartz-seq2 yielded primer-derived by-products completely overlapping in size with the those of cell hashing antibodies. Therefore, TAS-Seq may greatly enhance the tolerance and reduce the difficulties of TdT reactions, enhancing compatibility with DNA-barcoded antibody-based methods, including cell hashing and CITE-seq[35].

We found that TAS-Seq may outperform 10X v2 and Smart-seq2 in murine lung, kidney, and spleen tissues in terms of gene-detection sensitivity, highly variable gene detection sensitivity, and gene drop-out rates, indicating that TAS-Seq is the most sensitive of TdT-based scRNA-seq methods. Indeed, we showed that TAS-Seq detects genes across a wide range of expression levels more uniformly than 10X v2 and Smart-seq2, while possibly exhibiting more robust detection of growth factor and interleukin genes in murine lung. However, a possible drawback of TAS-Seq approach is that it limits theoretical tagging efficiency of second-strand synthesis reactions because short polyN-tailed cDNAs, which may not have sufficient polyN tails for annealing tagging primers, are stochastically produced. Further improvements to tagging primer structure for annealing short polyN-tailed cDNAs more efficiently are necessary to enhance TAS-Seq performance.

Clarifying cellular heterogeneity and composition is a major goal of scRNA-seq analysis. In murine lung tissue, TAS-Seq data showed higher a correlation with flow-cytometric data than 10X v2 and Smart-seq2 data, and also showed a high correlation with human lung data. In addition, TAS-Seq detected a similar number of human lung neutrophils as that found by flow-cytometry. Conversely, 10X v2 data showed high batch effects against cell composition in Tabula Muris lung data, and did not detect human lung neutrophils, and Smart-seq2 did not detect murine lung alveolar macrophages. Because Smart-seq2 data and 10X v2 P7-8/P7-9 data of murine lungs were obtained similarly by the Tabula Muris Consortium with minimal technical differences, Smart-seq2 may have yielded cell composition data better correlated with flow-cytometric data than that of 10X v2, and 10X v2 might have exhibited more batch effects than TAS-Seq. These characteristics may come from the difference in cell capture strategies; TAS-Seq used a BD Rhapsody nanowell system and cells were collected by gravity flow in isotonic buffer, representing milder isolation conditions for cells than the 10X system (cell suspension mixed with nuclease-free water changes osmotic pressure, possibly damaging cells) and Smart-seq2 (cells are isolated into microtiter plates by cell sorter that induces mechanical damage). The other possibility that might lead such a difference is that the BD Rhapsody system, but not the 10X system, can use a strong, detergent-based lysis buffer to efficiently inactivate

RNases, as the buffer is completely washed out before reverse transcription reaction. These characteristics of the nanowell system make TAS-Seq more suitable for analyzing fragile and RNase-rich cells than droplet or cell sorter-based scRNA-seq platforms. A possible reason for the accuracy of cell composition detection by TAS-Seq is the high sensitivity to long and poorly expressed genes compared to that of 10X v2 and Smart-seq2, which could lead efficient detection of cells with low mRNA levels. Further investigation of the sources of accuracy will be helpful for constructing a precise cell atlas by scRNA-seq analysis.

We explored intrinsic gene-detection bias in TAS-Seq, and found that genes detected only by TAS-Seq contained longer transcripts than those commonly detected by 10X v2 or Smart-seq2 in mouse lung cell subsets. Because template switching mainly occurred on the 5′ capped terminus of mRNA, the reverse transcription reaction must reach the 5′ termini for a successful template-switching reaction[7]. On the other hand, because TdT-based second-strand synthesis does not depend on 5′ terminal structure of mRNA, TdT-based scRNA-seq methods can capture cDNAs that are not elongated to 5′ termini. These differences may explain the discrepancy in transcript length distribution of detected genes between TdT-based TAS-Seq and template-switching-based 10X v2/Smart-seq2. We also found that TAS-Seq more favorably detected AT-rich genes than 10X v2 and Smart-seq2 in mouse lung cell subsets, unlike Quartz-seq, which detects GC-rich genes more favorably[15]. One of the major differences between TAS-Seq and Quartz-seq is that TAS-Seq uses polyC-tailing, whereas Quartz-seq uses polyA-tailing in the TdT reaction. In the TdT reaction process, short polyN-tailed cDNA given a hairpin structure by intra- or inter-molecular annealing of the polyN tail to the intermediate transcript region could be less efficiently extended by TdT because TdT-tailing efficiently is dropped down against the recessive end of double-stranded DNA[12]. The tagging primer for second-strand synthesis may less efficiently anneal and extend against such cDNAs than cDNAs with long polyN tail without intra- or inter-molecular annealing. Hence, it is possible that AT- and GC-rich genes, which are difficult to amplify via polyA- and polyC-based TdT-based cDNA amplification, may explain the AT- and GC-rich gene detection bias in TAS-Seq and Quartz-seq. Additives to prevent hairpin formation of single-stranded DNA may help to enhance the sensitivity of TdT-based scRNA-seq methods.

Cell–cell interaction analysis of TAS-Seq, 10X v2, and Smart-seq2 datasets revealed that TAS-Seq detected more interactions and important cell–cell communications for lung homeostasis in murine lung tissue than 10X v2 and Smart-seq2. When the threshold for the percentage of expressed cells of each ligand-receptor gene became more stringent, the number of inferred cell–cell interactions decreased in all methods. Thus, the lower drop-out rate of each gene in TAS-Seq than 10X v2 and Smart-seq2 data indicate the robustness of detection of cell–cell interactions. In addition, because CellChat analysis considered the abundance of each cell subset to calculate weights for each cell–cell interaction, the accuracy of cell composition in TAS-Seq data may be beneficial for the detection of important cell–cell interactions. Indeed, only TAS-Seq data identified strong interaction among AT2 cells, alveolar fibroblasts, and vascular endothelial cells in adult murine lung tissue, all of which are important in alveolar homeostasis, repair, and regeneration[25]. Further investigation of the performance of cell composition accuracy of various scRNA-seq platform might be important to construct cell–cell interaction atlas based on scRNA-seq data.

Our study did have some limitations. First, because we compared TAS-Seq data with publicly available scRNA-seq data, we could not exclude the effect cell preparation techniques. Conducting experiments in the same laboratory with the same

samples and many replicates will improve the evaluation accuracy of the performance comparison between TAS-Seq and other scRNA-seq methods. Second, we analyzed only mouse lung, spleen, kidney, and human lung tissues, and further investigation is necessary to evaluate the utility of TAS-Seq in other samples and biological conditions. Third, we did not perform extensive optimization of the TAS-Seq reaction conditions in this study as performed in the template-switching-based methods Smart-seq3[36] and FLASH-Seq[37]. Further investigation is necessary to optimize reaction conditions, enhance performance, and broaden the applications of TAS-Seq; options include modifications to buffer compositions, primer structures, and DNA polymerases for each solid DNA support.

Some scRNA-seq and spatial transcriptomics methods rely on solid-supported cDNA amplification. Drop-seq[38] (using polystyrene beads), Strereo-seq[39] (using DNBseq arrays), and 10X Visium™ are based on template switching, whereasSeq-scope[40] (using Illumina Miseq arrays) is based on a random priming approach. Of note, Seq-Well S³ and Slide-SeqV2[41,42] exhibit high sensitivity compared to their original methods, which only use template switching alone, by combining template switching and subsequent random priming approaches. This may be due to rescuing of cDNA molecules that are not rescued in a template-switching reaction. Because cDNA loss by washing out reaction components is minimal in solid-phase cDNA amplification methods, combining mechanistically different second-strand synthesis methods, including template-switching, TAS-Seq, and random priming, may enhance the gene-detection sensitivity of these solid phase–based scRNA-seq/spatial transcriptomics platforms.

Overall, TAS-Seq might be more easy-to-handle than existing TdT-based scRNA-seq methods and might possibly provide high-resolution scRNA-seq data with better accuracy of cell composition and inference of cell–cell interaction network than template-switching-based Smart-seq2 and 10X v2. Expanding TAS-Seq application is possibly helpful for better understanding and atlas construction of various biological contexts at the single-cell level.

## Methods

**Mice**. C57BL/6 J female mice were purchased from Sankyo Labo Service Corporation (Ibaragi, Japan). All mice were bred at specific pathogen-free facilities at Tokyo University of Science and were 8 weeks old (for lung sample) or 10 weeks old (for spleen sample) at the commencement of experiments.

**RNA extraction**. NIH/3T3 cells were cultured with DMEM high glucose (Nacalai Tesque, Kyoto, Japan) supplemented with 10% FBS (Cat#2916546, Lot#1608A, MP Bio Japan, Tokyo, Japan) and 10 mM HEPES (Cytiva (Global Life Sciences Technologies Japan), Tokyo, Japan) (DMEM/10%FBS/HEPES), and stored at −80 °C with CellBanker 1 (Zenoaq Resource, Fukushima, Japan). Stored cells were thawed and cultured with DMEM/10%FBS/HEPES, and 80% of confluent cells were recovered. Total RNA was extracted from resultant cells using TRIZol Reagent (Thermo Fisher Scientific, Tokyo, Japan) according to the manufacturer's instructions. The extracted RNA was dissolved with nuclease-free water (Nacalai Tesque) and stored at −80 °C.

**Preparation of cDNA-immobilized or un-immobilized BD Rhapsody beads for evaluation of TAS-Seq**. BD Rhapsody magnetic beads (BD Biosciences, San Jose, CA, USA) for bulk experiments were collected as follows. Un-trapped BD Rhapsody beads after being loaded onto BD Rhapsody cartridge were collected, washed twice with WTA wash buffer [10 mM Tris-HCl pH 8.0 (Nippon Gene, Tokyo, Japan), 50 mM NaCl (Merck, Tokyo, Japan), 1 mM EDTA (Nippon Gene, Tokyo, Japan), and 0.05% Tween-20 (Merck)], were resuspended with 200 μL of Beads resuspension buffer (BD Biosciences) and stored at 4 °C. After removing the supernatant, 1 μg of total RNA from NIH/3T3 cells was diluted with 500 μL of BD Rhapsody Lysis buffer (BD Biosciences) supplemented with 15 mM DTT (BD Biosciences) and added to the beads. Beads were resuspended and incubated for 30 min at room temperature (RT) with gentle rotation. Beads were washed once with 500 μL of BD Rhapsody lysis buffer, once with 1 ml of wash buffer B [10 mM Tris-HCl (pH 7.5) (Nippon Gene), 150 mM LiCl (Merck), 1 mM EDTA, and 0.02% Tween-20], and twice with 500 μL of wash buffer B. During the washing step, bead-

containing DNA LoBind tubes (Eppendorf Japan, Tokyo, Japan) were replaced twice. After removing the supernatant, reverse transcription was performed for 20 min at 37 °C using a BD Rhapsody cDNA kit following the manufacturer's instruction. After removing the supernatant, Exonuclease I mix (20 μL of 10X Exonuclease I buffer (New England Biolabs, Ipswich, MA, USA), 10 μL of Exonuclease I (New England Biolabs), and 170 μL of nuclease-free water (Nacalai Tesque) in 200 μL reaction) was directly added to the beads and further incubated for 60 min at 37 °C with 1,200 rpm on a Thermomixer C with Thermotop (Eppendorf Japan). Resultant beads were immediately chilled on ice, the supernatant was removed, and washed with 1 mL of WTA wash buffer, 200 μL of BD Rhapsody lysis buffer (for inactivation of enzyme), once with 1 ml of WTA wash buffer, and twice with 500 μL of WTA wash buffer, resuspended with 200 μL of Beads resuspension buffer (BD Biosciences) and stored at 4 °C. During the washing step, bead-containing DNA LoBind tubes (Eppendorf Japan, Tokyo, Japan) were replaced twice. For producing cDNA un-immobilized BD Rhapsody beads, un-trapped BD Rhapsody beads were purchased as above. After removing the supernatant, beads were treated with Exonuclease I mix (20 μL of 10X Exonuclease I buffer, 10 μL of Exonuclease I, and 170 μL of nuclease-free water in 200 μL reaction) for 60 min at 37 °C with 1,200 rpm on a Thermomixer C with Thermotop. Resultant beads were immediately chilled on ice; the supernatant was removed and washed with 1 mL of WTA wash buffer, 200 μL of BD Rhapsody lysis buffer (for inactivation of enzyme), once with 1 ml of WTA wash buffer, twice with 500 μL of WTA wash buffer, resuspended with 200 μL of Beads resuspension buffer and stored at 4 °C. During the washing step, bead-containing DNA LoBind tubes were replaced twice. For washing BD Rhapsody beads, BD IMagnet Cell Separation Magnet (BD Biosciences) and Dynamag-2 (Thermo Fisher Scientific) were used for collecting BD Rhapsody beads.

**Evaluation of terminator-assisted homopolymer tailing reaction and DNA amplification from BD Rhapsody beads**. For cDNA un-immobilized beads, beads were split into seven parts, transferred into 1.5 ml DNA LoBind tubes, and subjected to homopolymer tailing reaction by terminal transferase (TdT). After removing the supernatant and washing once with nuclease-free water, the beads were mixed with TdT mixture 1 [1×TdT buffer (Thermo Fisher Scientific), 1.2 mM deoxycytidine triphosphate (dCTP, Roche Diagnostics, Tokyo, Japan), 0.06 mM dideoxycytidine triphosphate (ddCTP, Cytiva), 15 U/μL TdT (Roche), 0.1 U/μL RNase H (QIAGEN, Düsseldorf, Germany)], TdT mixture 2 [1 × TdT buffer, 1.2 mM dCTP, 0.06 mM ddCTP, 10 U/μL TdT, 0.1 U/μL RNase H], TdT mixture 3 [1 × TdT buffer, 1.2 mM dCTP, 0.06 mM ddCTP, 42 U/μL TdT, 0.1 U/μL RNase H], TdT mixture 4 [1 × TdT buffer, 1.2 mM dCTP, 0.06 mM ddCTP, 15 U/μL TdT, 0.1 U/μL RNase H], and no TdT control mixture [1 × TdT buffer, 1.2 mM dCTP, 0.06 mM ddCTP, 0.1 U/μL RNase H]. TdT reactions were performed using 100 μL/tubes for 5 or 30 min (TdT mixture 1 and 4) and for 30 min (TdT mixture 2, 3, and no TdT control mixture) at 37 °C with 1,200 rpm on a Thermomixer C with Thermotop. For cDNA-immobilized beads, beads were split into four parts, transferred into 1.5 ml DNA LoBind tubes. After removing the supernatant and washing once with nuclease-free water, the three parts of the beads were mixed with the TdT mixture 1, and one was mixed with the no TdT control mixture. Then, beads were incubated for 15, 30, and 45 min (TdT mixture 1) and 45 min (no TdT control mixture) at 37 °C with 1,200 rpm on a Thermomixer C with Thermotop. Reactions were chilled on ice immediately after the reaction was completed. After the supernatant was removed, beads were washed with 1 mL of WTA wash buffer, 200 μL of BD Rhapsody lysis buffer (for inactivation of enzyme), once with 1 ml of WTA wash buffer, twice with 500 μL of WTA wash buffer, and resuspended with 100 μL of 10 mM Tris-HCl pH 8.0. During the washing step, bead-containing DNA LoBind tubes were replaced twice. Beads were transferred into new 8-strip tubes, the supernatant was discarded, and 12.5 μL of second-strand synthesis mixture [1× KAPA Hifi ReadyMix (KAPA Biosystems, Wilmington, MA, USA) and 0.4 μM 5'universal-9G primer] was added, and second-strand synthesis was performed according to the following program: 95 °C for 3 min, 98 °C for 20 s, 47 °C for 2 min, 72 °C for 7 min, and hold at 4 °C. Then, 37.5 μL of amplification mix [1× KAPA Hifi ReadyMix, 0.4 μM 3' universal primer, and 0.267 μM 5' universal primer] was added and PCR performed using the following program: 95 °C for 3 min, 7 cycles (for no cDNA-immobilized beads) or 9 cycles (for cDNA-immobilized beads) of 98 °C for 20 s, 63 °C for 20 s, and 72 °C for 5 min followed by 72 °C for 5 min and hold at 4 °C. PCR products were purified once with a 3.0× Pronex size-selective purification system (Promega, Madison, WI, USA) and eluted with 22 μL of 10 mM Tris-HCl pH 8.0. Amplified products were quantified using a Nanodrop 8000 (Thermo Fisher Scientific), and size distribution was analyzed by Agilent High Sensitivity DNA kit (Agilent Technologies, Santa Clara, CA, USA) with Agilent 2100 Bioanalyzer (Agilent Technologies) with appropriate dilutions.

**Assessment of the extension of remaining primer-derived products in the Quartz-seq2 protocol**. We assessed the extension of remaining primer-derived products in the Quartz-seq2 protocol[14] according to the previous report, without any RNA. Briefly, the reverse transcription reaction was performed at a total volume of 700 μL and contained Quartz-seq2 MDRT001 primer without any RNA addition. This was purified with a Zymo Clean&Concentrator-5 column (Zymo Research, CA, USA) and eluted twice with 10.5 μL of nuclease-free H₂O. The TdT reaction was prepared using 4 μL of the resulting elutant and 5 μL of ice-cold TdT

mix (standard Quartz-seq2 condition: 1× Thermopol buffer [New England Bio-labs], 0.24 mM dATP [Invitrogen], 0.0384 U/μL RNase H [Invitrogen], 26.88 U/μL TdT [Roche]; 1.5× TdT Quartz-seq2 condition: 1× Thermopol buffer, 0.24 mM dATP, 0.0384 U/μL RNase H, 40.32U/μL TdT). To simulate the accidental warming of the TdT reaction, a subset of the reaction was incubated at 23 °C for 5, 10, or 15 min and immediately chilled on ice. The TdT reaction was performed using the following program: 0 °C for 5 sec, 37 °C for 75 sec, 65 °C for 10 min, hold at 4 °C. Second-strand synthesis and PCR amplification reactions were performed according to the Quartz-seq2 protocol with a total of 15 PCR cycles. The reaction was purified with Zymo clean&concentrator-5 column and 1.8× AmPure XP beads. Size distributions of the resulting reactions were analyzed using a MultiNA system (Shimazu, Kyoto, Japan).

**Single-cell preparation.** Lung cells were prepared as described previously[43] with some modifications. Briefly, mice were anesthetized with isoflurane, lungs were perfused with PBS (Nacalai Tesque), and the left lung was collected. Human lung samples were collected from lung cancer patients with pulmonary fibrosis who underwent curative surgical resection from August 2015 to December 2019 at Nara Medical University Hospital. Informed consent was obtained from all patients who participated in the study herein. Removed lung samples were determined as non-fibrosis and fibrosis areas without lung cancer under thin-section computed tomography by two independent respiratory specialists. Murine or human lung samples were minced into 0.5 mm$^2$ with a razor blade and digested with Liberase solution [RPMI-1640 (Nacalai Tesque) supplemented with 10% FBS, 10 mM HEPES pH 7.2–7.4, 0.25 mg/ml Liberase TM (Roche), and 2 kU/mL DNase I (Merck)] at 37 °C for 60 min. For murine samples, the cell suspension was agitated 20 times with an 18 G needle (Terumo, Tokyo, Japan) after 20 min incubation, agitated 20 times with 21 G needle (Terumo) after 40 min incubation, and agitated 10 times with 200 μL pipette tip. For human samples, the cell suspension was agitated 20 times with an 18 G needle (Terumo) every 20 min incubation. Cell suspensions were passed through a 70 μm cell strainer (BD Biosciences), centrifuged at 4 °C for 500 × g for 5 min, and their supernatant was discarded. Cells were resuspended with 25% Percoll PLUS (Cytiva), agitated with an 18 G needle five times. After under layered 65% Percoll PLUS, cell suspensions were centrifuged at 20 °C for 1000 × g for 20 min, and the middle layer was collected. Resultant cell suspensions were diluted thrice with preparation medium [RPMI-1640 supplemented with 5% FBS and 10 mM HEPES], centrifuged 500 × g at 4 °C for 7 min, and their supernatants were discarded. Resultant cells were resuspended with 500 μL of preparation medium. Tumor cells of the subcutaneous model of lewis lung carcinoma were collected as described previously[18]. Spleen cells were collected as follows. The spleen tissue from 8-week-old B6 mice was mushed on a cell strainer with 5 mL of ACK lysis buffer [155 mM NH$_4$Cl, 10 mM KHCO$_3$, 0.1 mM EDTA, pH 7.3] and incubated for 1 min at 20 °C. Then, 5 mL of preparation medium were added and centrifuged at 4 °C for 500 × g for 5 min, suspended in 1 mL of ACK lysis buffer, and incubated for 1 min at 20 °C. The cell suspension was washed twice with 5 mL of preparation medium and subjected to BD Rhapsody analysis. Kidney cells were collected as follows. Murine kidney lung samples were minced into 0.5 mm$^2$ with a razor blade and digested with Liberase solution [RPMI-1640 (Nacalai Tesque) supplemented with 10% FBS, 10 mM HEPES pH 7.2–7.4, 0.5 mg/ml Liberase TH (Roche), and 2 kU/mL DNase I (Merck)] at 37 °C for 60 min. The cell suspension was agitated 20 times with an 18 G needle (Terumo, Tokyo, Japan) after 20 min incubation, agitated 20 times with 21 G needle (Terumo) after 40 min incubation, and agitated 10 times with 200 μL pipette tip. Cell suspensions were passed through a 70 μm cell strainer (BD Biosciences), centrifuged at 4 °C for 500 × g for 5 min, and their supernatant was discarded. Cells were resuspended with 25% Percoll PLUS (Cytiva), agitated with an 18 G needle five times. After under layered 65% Percoll PLUS, cell suspensions were centrifuged at 20 °C for 1000 × g for 20 min, and the middle layer was collected. Resultant cell suspensions were diluted thrice with preparation medium [RPMI-1640 supplemented with 5% FBS and 10 mM HEPES], centrifuged 500 × g at 4 °C for 7 min, and their supernatants were discarded. Resultant cells were resuspended with 500 μL of preparation medium. Each single-cell suspension cell concentration was counted using Flow-count fluorospheres (Beckman Coulter, Brea, CA, USA) and a Gallios flow cytometer (Beckman Coulter) or a CytoFLEX S flow cytometer (Beckman Coulter).

**Flow cytometry.** For murine lung cells, single-cell suspensions were blocked with Fc block (anti-CD16/32, clone: 2.4G2, BioXcell, West Lebanon, NH, USA) and True-Stain Monocyte Blocker (BioLegend), then stained with appropriate antibody mixtures diluted with PBS supplemented with 2% FBS (2% FBS/PBS). After washing with PBS supplemented with 2% FBS/PBS, cells were suspended with 2% FBS/PBS and 0.5 μg/ml propidium iodide. For human lung cells, single-cell suspensions were washed once with PBS and stained with LIVE/DEAD Fixable Aqua Dead Cell Stain Kit (Thermo Fisher Scientific) at 4 °C for 30 min. Cells were washed once with 2% FBS/PBS; cells were blocked with 2% normal mouse serum and stained with appropriate antibody mixtures diluted with 2% FBS/PBS. After washing with 2% FBS/PBS, cells were suspended with 2% FBS/PBS. Data were collected on a Gallios flow cytometer or a CytoFLEX S flow cytometer and analyzed using FlowJo software v10.6.2 (BD Biosciences). A detailed list of used antibodies is shown in Supplementary Data 2.

**cDNA synthesis and Exonuclease I treatment by BD Rhapsody system.** For cell hashing, CD45$^+$ tumor-infiltrating leukocytes were stained with 2.5 μg/ml of Totalseq anti-mouse Hashtag-A antibodies (A0301-A0314, BioLegend, San Diego, CA, USA) at 4 °C for 25 min and washed thrice with Cell Staining Buffer (Bio-Legend) and pooled equally as described previously[18]. Obtained single-cell suspensions were subjected to a BD Rhapsody system with BD Rhapsody Targeted & Abseq Reagent kit (BD Biosciences) following the manufacturer's instructions. 20000 mouse lung cells, 20000 mouse CD45$^+$ tumor-infiltrating leukocytes, and 20000 mouse spleen cells were subjected to the BD Rhapsody workflow, and 5000 mouse kidney cells and 6000 human lung cells were subjected to the BD Rhapsody Express workflow. After reverse transcription, Exonuclease I treatment of the resultant BD Rhapsody beads was performed at 37 °C for 60 min with 1,200 rpm on a Thermomixer C with Thermotop. Resultant beads were immediately chilled on ice; the supernatant was removed and washed with 1 mL of WTA wash buffer, 200 μL of BD Rhapsody lysis buffer (for inactivation of enzyme), once with 1 ml of WTA wash buffer, twice with 500 μL of WTA wash buffer, resuspended with 200 μL of Beads resuspension buffer and stored at 4 °C. During the washing step, bead-containing DNA LoBind tubes were replaced twice. For the spleen cell sample subjected to BD WTA kit (BD Biosciences), half of the BD Rhapsody beads were split just after Exonuclease I treatment, and the enzyme was heat-inactivated at 80 °C for 20 min.

**Amplification of cDNA by BD Rhapsody WTA kit.** Half of the Exonuclease I-treated BD Rhapsody beads from the spleen cell sample were subjected to BD Rhapsody kit for cDNA amplification following manufacturer instructions.

**Amplification of cDNA and hashtag libraries by TAS-Seq from BD Rhapsody beads.** The reverse-transcribed, Exonuclease I-treated BD Rhapsody beads were subjected to TAS-Seq workflow for cDNA and/or hashtag library amplification. After removing the supernatant and washing once with nuclease-free water, the beads were mixed with 200 μL of TdT mixture [1× TdT buffer, 1.2 mM dCTP, 0.06 mM ddCTP, 15 U/μL TdT, 0.1 U/μL RNase H] and incubated for 30 min at 37 °C with 1200 rpm on a Thermomixer C with Thermotop. Reactions were chilled on ice immediately after the reaction was completed. After the supernatant was removed, beads were washed with 1 mL of WTA wash buffer, 200 μL of BD Rhapsody lysis buffer, once with 1 ml of WTA wash buffer, twice with 500 μL of WTA wash buffer, and resuspended with 100 μL of 10 mM Tris-HCl pH 8.0. During the washing step, bead-containing DNA LoBind tubes were replaced twice. Beads were split into two parts and transferred into new 8-strip tubes, the supernatant was discarded, and 25 μL of second-strand synthesis mixture (1× KAPA Hifi ReadyMix and 0.4 μM 5' WTA-9G primer [for mouse lung cells, spleen cells, and tumor-infiltrating leukocytes] or 5' LibA-9G primer ([for human lung samples, mouse kidney sample, and mouse lung TAS-Seq dataset2 sample])) was added, and second-strand synthesis was performed according to the following program: 95 °C for 3 min, 98 °C for 20 s, 47 °C for 2 min, 72 °C for 7 min, and hold at 4 °C. Then, 75 μL of 1st round of whole-transcriptome amplification (WTA) mix [1× KAPA Hifi ReadyMix, 0.4 μM 3' universal primer, and 0.267 μM 5' WTA primer] (for mouse lung and spleen cells), [1× KAPA Hifi ReadyMix, 0.4 μM 3' universal primer, and 0.267 μM 5' LibA primer] (for mouse lung TAS-Seq dataset2 samples and human lung cells), or [1× KAPA Hifi ReadyMix, 0.4 μM 3' universal primer, 0.267 μM 5' universal primer, and 0.267 μM 5' hashtag primer] (for tumor-infiltrating leukocytes) was added, split samples into two tubes (50 μL each), and PCR performed using the following program: 95 °C for 3 min, seven cycles of 98 °C for 20 s, 63 °C for 20 s, and 72 °C for 5 min, followed by 72 °C for 5 min and hold at 4 °C. PCR products with no hashtag libraries (spleen, mouse lung, and human lung cells) were combined and purified twice with 0.65× AmPure XP beads (Beckman Coulter) and eluted with 21 μL of nuclease-free water. PCR products with hashtag libraries (for tumor-infiltrating leukocytes) were combined, and cDNA product was purified by 0.65× AmPure XP beads, and unbounded fraction was isolated. Hashtag product was purified from the unbounded fraction by adding additional 0.7× AmPure XP beads (final 1.35×). Then, cDNA and hashtag libraries were further purified by 0.65× and 1.35 × AmPure XP beads, respectively, and eluted with 21 μL of nuclease-free water. For amplification of the cDNA libraries, 2nd round of WTA mix [25 μL of 2× KAPA Hifi ReadyMix, 2 μL of 10 μM 3' universal primer, and 2 μL of 10 μM 5' WTA primer (for spleen cells and tumor-infiltrating leukocytes) or 5' LibA primer (for human and mouse lung samples)] was added to the cDNA libraries, and PCR performed using the following program: 95 °C for 3 min, 9 cycles (for mouse and human lung cells) or 11 cycles (for spleen cells) of 98 °C for 20 s, 63 °C for 20 s, and 72 °C for 5 min followed by 72 °C for 5 min and hold at 4 °C. For amplification of the hashtag libraries, 2nd round of hashtag-amplification mix [25 μL of 2× KAPA Hifi ReadyMix, 2 μL of 10 μM 3' universal primer, and 2 μL of 10 μM 5'hashtag primer] was added to the cDNA libraries, and PCR performed using the following program: 95 °C for 3 min, 9 cycles of 98 °C for 20 s, 63 °C for 20 s, and 72 °C for 45 sec, followed by 72 °C for 5 min and hold at 4 °C. Amplified products were purified two times with 0.65× AmPure XP beads (for cDNA libraries) or 1.35× AmPure XP beads (hashtag libraries) and eluted with 30 μL of 10 mM Tris-HCl pH 8.0. Then, barcoded PCR mix for hashtag library [1× KAPA Hifi ReadyMix, 0.4 μM 3' i5-UDI0033 primer, 0.4 μM i7-UDI0033 primer, 5 ng of purified hashtag library] was prepared and PCR performed using the following program: 95 °C for 3 min, 9

cycles of 98 °C for 20 s, 63 °C for 20 s, and 72 °C for 45 sec, followed by 72 °C for 5 min and hold at 4 °C. Amplified products were purified by double size selection with 0.8× → 0.4× (final 1.2×) AmPure XP beads and eluted with 25 μL of 10 mM Tris-HCl pH8.0. Amplified products were quantified using a Nanodrop 8000, and size distribution was analyzed by Agilent High Sensitivity DNA kit with an Agilent 2100 Bioanalyzer or a MultiNA system (Shimazu, Kyoto, Japan) with appropriate dilutions. Primer sequences used for this study were shown in Supplementary Data 3.

**Illumina library construction and sequencing.** Illumina libraries were constructed from 100 ng of amplified cDNA libraries using the NEBNext Ultra II FS library prep kit for Illumina (New England Biolabs) with some modifications. Briefly, fragmentation, end-repair, and A-tailing were performed using the following program: 32 °C for 5 min, 65 °C for 30 min, and hold at 4 °C. Then, 2.5 μL of 3.3 μM illumine adapter was used for adapter ligation. Ligated products were purified by double size selection with 10 μL → 25 μL AmPure XP beads and eluted with 15 μL of nuclease-free water. Nine cycles of Barcoding PCR were performed using i5-UDI00XX and i7-UDI00XX primers. Resultant products were purified twice by double size selection with 0.5× → 0.3× (final 0.8×) AmPure XP beads and eluted with 30 μL of 10 mM Tris-HCl pH8.0. Size distribution of amplified products was analyzed by Agilent High Sensitivity DNA kit with Agilent 2100 Bioanalyzer or MultiNA system (Shimazu, Kyoto, Japan) with appropriate dilutions. Resultant libraries and barcoded hashtag libraries were quantified using the KAPA Library Quantification Kit (KAPA Biosystems). Primer sequences used for this study were shown in Supplementary Data 3. The primers were purchased from Eurofins Genomics (Tokyo, Japan) or Integrated DNA Technologies (Coralville, IA USA). Sequencing was performed by Illumina Novaseq 6000 sequencer (Illumina, San Diego, CA, USA) and NovaSeq 6000 S4 Reagent Kit v1.5 (200 cycles) or NovaSeq 6000 S2 Reagent Kit v1.0 (100 cycles) (Illumina) following manufacturer instructions. Pooled library concentration was adjusted to 2.0 nM (v1.5 kit) or 1.75 nM (v1.0 kit), and 12% PhiX control library v3 (Illumina) was spiked into the library. Sequencing configurations were follows: murine lung and spleen datasets of TAS-Seq/BD WTA kit (NovaSeq 6000 S4 Reagent Kit v1.5 [200 cycles], read1 67 base-pair [bp], read2 151 bp, index1 8 bp, index2 8 bp); murine lung dataset2 of TAS-Seq (NovaSeq 6000 S2 Reagent Kit v1.0 [200 cycles], read1 101 bp, read2 101 bp, index1 8 bp, index2 8 bp); murine kidney and human lung datasets (NovaSeq 6000 S2 Reagent Kit v1.0 (100 cycles), read1 61 bp, read2 50 bp, index1 6 bp, index2 6 bp).

**Fastq data preprocessing and generation of the single-cell gene-expression matrix.** Pair-end Fastq files (R1: cell barcode reads, R2: RNA reads) of TAS-Seq and BD WTA kit data were processed as follows. Adapter trimming of sequencing data was performed using cutadapt 3.4[44]. Filtered reads were chunked into 16 parts for parallel processing by using Seqkit 0.15.0[45]. Filtered cell barcode reads were annotated by Python script provided by BD Biosciences with minor modification for compatibility to Python 3.7. Reference RNA sequences were built by concatenating cDNA and ncRNA fasta files of the Ensembl database (build GRCm38 release-101 for mouse data and GRCh38 release-101 for human data)[46]. Associated cDNA reads were mapped to reference RNA using bowtie2-2.4.2[47] by the following parameters: -p 2 -very-sensitive-local -N 1 -norc -seed 656565 -reorder. Then, cell barcode information of each read was added to the bowtie2-mapped BAM files by the python script and pysam 0.15.4 (https://github.com/pysam-developers/pysam), and read counts of each gene in each cell barcode were counted using mawk. Resultant count data was converted to a single-cell gene-expression matrix file. The inflection point of the knee-plot (total read count versus the rank of the read count) was detected using DropletUtils package[48] in R-4.1.2 (https://cran.r-project.org/). Cells of which total read count was over inflection point were considered as valid cells. Because unique-molecule identifiers (UMIs) of BD Rhapsody beads are 8-base UMIs before polyT stretch, which might not be sufficient to exert theoretical UMI diversity by the distortion of base frequencies and to avoid UMI collision (more than 10-base UMIs is necessary for scRNA-seq datasets)[49], we did not use BD Rhapsody UMIs for TAS-Seq data and BD WTA data. Fastq files of 10X v2 datasets (GEO accession: GSM3926540, GSM3040906, GSM3040907, GSM3040896, and GSM3040903) were downloaded and re-analyzed using CellRanger-6.1.2 (10X Genomics, CA, USA). A non-UMI raw count matrix of 10X v2 datasets was obtained from the mapped and cell barcode–annotated BAM files of CellRanger output by the python script, pysam 0.15.4, and mawk. Resultant count data was converted to a single-cell gene-expression matrix file and filtered by valid cell barcode originally identified by CellRanger. Downsampled fastq files were obtained by using split2 command of Seqkit 0.15.0.

**Background subtraction of TAS-Seq expression matrix by distribution-based error correction (DBEC).** To reduce background read counts of each gene that were possibly derived from RNA diffusion during cell lysis step within BD Rhapsody cartridge and reverse transcription, we performed distribution-based error correction (DBEC) that is included in BD Rhapsody targeted scRNA-seq workflow. To estimate background and signal read count distribution, we used the Gaussian mixture model previously used to estimate the gene-expression distribution of scRNA-seq datasets[50]. First, genes of which $\log_2(x+1)$-transformed

maximum expression over 8 were selected, and biexponential transformation was applied to each gene count by using FlowTrans package[51] in R-4.1.2. Next, Gaussian mixture components (model E, from one to three components) were detected using mclust package[52] in R-4.1.2, and the average expression of each component was calculated. Genes of which the maximum average expression of each component was over 5.5 (for shallow-sequenced data) or over 6 (for deep-sequenced data) were selected. Then, if the difference of the average expression of each component against their maximum expression was greater than 5 (for shallow-sequenced data) or over 5.5 (for deep-sequenced data), the expression level of the components was considered to be background gene expression and converted expression of the components to 0.

**Single-cell clustering and annotation.** Clustering of single cells of each dataset was performed using Seurat v4.0.3[19] in R-4.1.2. For Tabula Muris Smart-seq2 data, the expression value of ERCC spike-ins was excluded. Seurat object for each dataset was created using CreateSeuratObject function (min.cells = 5, min.genes = 500). scRNA-seq library metrics, including mitochondrial gene-count proportion, ribosomal protein gene-count proportion, and ribosomal RNA count proportion, were calculated using R-4.1.2 and visualized using geom_hex (bins = 100) function in ggplot2 package[53]. Gene count and $\log_{10}$ converted read count distribution was visualized using the RidgePlot function in the Seurat package with default parameters. Cells of which mitochondrial gene proportion was over 0.25 were filtered out by subset function in Seurat v4.0.3. The expression data was normalized by normalizeData function (scale-factor = 1,000,000 (according to the analytical parameter used by Tabula Muris[4])), and scaled with ScaleData function in Seurat v4.0.3. Read counts of each cell within each dataset were regressed as a confounding factor within the ScaleData function. Highly variable genes of each dataset were identified using the FindVariableFeatures function in Seurat v4.0.3. with the following parameters: selection.method = "mvp", mean.cut-off = c(0.1, Inf), dispersion.cutoff = c(0.5, Inf). Then, principal component analysis (PCA) against identified highly variable genes and projection of PCA onto entire data was performed using RunPCA (number of calculated PCs were 150) function in Seurat v4.0.3. Enrichment of each PC was calculated using the JackStraw function (num.replicate = 100), and PCs that were significantly enriched statistically ($p \leq 1 \times 10^{-5}$) were selected for clustering and dimensional reduction analyses. Cell clustering was performed using FindClusters function (resolution = 1.5 (for Smart-seq2 data, 10X v2 data of GSM3926540, and TAS-Seq data of murine lung shallow-sequenced data), 1.0 (for 10X v2 data of Tabula Muris), 3.0 (TAS-Seq data of murine lung deep-sequenced data), 3.5 (TAS-Seq dataset2 of murine lung), 2.5 (for TAS-Seq data of human lungs)) in Seurat v4.0.3 against the significant PCs, and dimensional reduction was performed using python wrapper of Fast Fourier transform-accelerated interpolation-based t-stochastic neighbor embedding (FIt-SNE)[54] v1.2.1 (perplexity = 100, df = 0.9, random_seed = 42, max_iter = 1000, and all the other parameters were set as defaults) through reticulate package (https://github.com/rstudio/reticulate) in R-4.1.2. Statistically significant marker genes of each identified cluster were identified using parallelized FindMarkers function in Seurat v4.0.3 (test.use = "wilcox", only.pos = TRUE, min.pct = 0.1, logfc.threshold = 0.25, adjusted $p$ (Bonferroni correction) ≤ 0.05). Then, each identified cluster was manually annotated by their marker genes previously reported as cell subset-defining marker genes, and the lineage marker double-positive cells were annotated as doublets. Next, we further sub-clustered cell subsets that were not fully separated into known cell subsets (dendritic cell (DC) and T cell subsets of Tabula Muris Smart-seq2 and 10X v2 datasets, T cell subsets of 10X v2 GSM3926540 datasets, monocyte/DC/interstitial macrophage subset in TAS-Seq datasets) using Seurat v4.0.3 by the similar workflow of whole-cell data, and incorporate their annotation into Seurat object of whole-cell data (detail analysis parameters and associated codes were deposited at https://github.com/s-shichino1989/ TASSeq-paper.). Cell subset annotations and their compositions were visualized in 2D FIt-SNE space and stacking plot, respectively. All of the identified marker genes are shown in Supplementary Data 4, and cell cluster annotations are shown in Supplementary Data 1.

**Cell composition correlation analysis between flow-cytometric data and scRNA-seq data.** The percentage of the abundance of specific cell subsets against total cells were calculated, and Pearson's correlation coefficients, linear regression, slope of regression line, and associated $p$-values between the cell composition of flow-cytometric data and each scRNA-seq data were calculated using GraphPad Prism 9.1.2 (Graphpad Software, La Jolla, CA, USA).

**Comparison of drop-out rates, percent GC content, and transcript length of detected genes in each cell population.** Differences in drop-out rate of each detected gene of the commonly detected cell subset were calculated as follows: (percent of positive cells in TAS-Seq data) – (percent of positive cells in 10X v2 or Smart-seq2 data). Average expression of each gene in each cell subset was calculated using the normalized expression value of each cell. To compare the percentage of GC content and transcript length of each gene, genes for which (percent of positive cells in TAS-Seq data) − (percent of positive cells in 10X v2 or Smart-seq2 data) was ≥0.10 and genes of which was only detected by TAS-Seq were defined as highly detected genes by TAS-Seq. Genes for which (percent of positive cells in TAS-Seq data) − (percent of positive cells in 10X v2 or Smart-seq2 data) was ≤−0.10 and genes of which was only detected by 10X v2 or Smart-seq2 data were defined as highly detected genes by 10X v2 or Smart-seq2 data. Genes for which

absorbance ((percent of positive cells in TAS-Seq data) − (percent of positive cells in 10X v2 or Smart-seq2 data)) < 0.10 were defined as commonly detected genes. Information regarding the percentage of GC content and transcript length of each gene was extracted using the biomaRt package[55] in R-4.1.2.

**Calculation of gene-read distribution in Pseudo-bulk and bulk RNA-seq data**. Pseudo-bulk gene-expression matrices of each murine lung cell subset for TAS-Seq, 10X v2, and Smart-seq2 datasets were generated by summation of raw read counts. Then, the total read counts of pseudo-bulk data was normalized to 10,000,000 reads. For bulk RNA-seq data of sorted adult murine lung fibroblasts, we used our previously published datasets (GSE110540) and normalized by total read counts as 10,000,000 reads. The gene-read distribution of each lung cell subset was visualized by violin/box plot using the ggplot2 package. The Kullback-Leibler divergence of the gene-read distribution for lung fibroblasts among the datasets was calculated and visualized using the philentropy[56] and pheatmap packages in R-4.1.2.

**Inference of cell–cell interaction network from scRNA-seq data**. Inference of cell–cell interaction network of each murine lung scRNA-seq dataset was performed using CellChat v1.1.3 package[23]. First, total cell number was downsampled to 1717 cells (the total cell number of Smart-seq2 dataset) using the subset function of Seurat v4.0.3, re-normalized using NormalizeData function (scale.factor = 1,000,000) of Seurat v4.0.3., and CellChat objects of each dataset were created from raw expression data, normalized expression data, and associated cell-annotation metadata extracted from the Seurat objects. Identification of overexpressed interactions, calculation of communication probability between cell subsets, and identification of overexpressed pathways was performed according to the CellChat default workflow (https://github.com/sqjin/CellChat) changing the threshold of ligand-receptor gene-expression abundancy within cell subsets (changing thresh.pc parameter = 0.05, 0.15, 0.25, 0.5, and 0.75 in the identifyOverExpressedGenes function in CellChat v1.1.3). Outgoing/incoming signaling strength was calculated and visualized using the netAnalysis_signalingRole_scatter function in CellChat v1.1.3. Circle plot visualization of cell–cell interaction network and the strength of the communication between each cell subset were performed using the netVisual_circle function in CellChat v1.1.3. To visualize the whole cell–cell interaction network, the top 10% of interactions were selected using the netVisual_circle function. The compareInteractions function in CellChat v1.1.3 calculated the number of inferred cell–cell interactions of each dataset after merging CellChat objects. Heatmap visualization of identified cell–cell interaction pathways was performed using pheatmap package in R-4.1.2.

**Visualization of growth factor and interleukin gene expression patterns in murine lung cell subsets**. Expression patterns of selected growth factor and interleukin genes in each murine lung dataset were visualized using the DotPlot function in Seurat v4.0.3 in R-4.1.2. Cell subsets with rates of positive expression under 5% were filtered out from the DotPlot visualization.

**Statistics and reproducibility**. The significance of the difference of read count number among datasets, detected gene number among datasets, percent GC content, average expression, and transcript length between TAS-Seq-only genes and commonly detected genes were calculated using the wilcox_test function of rstatx package with holm's multiple correction in R-4.1.2. The significance of the enrichment of PCs and marker genes of each cell cluster was calculated using Seurat v4.0.3 package in R-4.1.2. The significance of the correlation between the cell composition of flow-cytometric data and each scRNA-seq data was calculated using GraphPad Prism 9.1.2 (Graphpad Software, La Jolla, CA, USA). All statistical analyses were conducted with a significance level of $\alpha = 0.05$ ($p \le 0.05$). Sample sizes and replicates are described in each figure legend.

**Study approval**. All animal experiments were reviewed and approved by the Animal Experiment Committee of Tokyo University of Science (approval number: S17034, S18029, S19024, and S20019). All human studies were approved by the Ethics Committee of Nara Medical University (Approval No. 1973) and Tokyo University of Science (Approval No. 18018).

**Reporting summary**. Further information on research design is available in the Nature Research Reporting Summary linked to this article.

## Data availability

Raw data, annotated gene-expression matrix, and associated metadata from these experiments have been deposited in the NCBI gene expression omnibus (GEO); accession GSE180149 and GSE200090. Public data used for this study is available at https://figshare.com/projects/Tabula_Muris_ Transcriptomic_characterization_of_20_organs_and_tissues_from_Mus_musculus_at_-single_cell_resolution/27733 (Tabula Muris Smart-seq2 data), GSE109774 (Tabula Muris 10X v2 data), and GSM3926540 (10X v2 shallow-sequenced murine lung data), https://www.nature.com/articles/s41586-020-2922-4#Sec33; Supplementary Data 2 (cell

abundancy data of human lungs of 10X v2 and Smart-seq2 data). All concentrations of antibodies and oligo sequences are shown in Supplementary Data 2 and Supplementary Data 3. Figures 3b–e, 4b, f were generated by R code and uploaded in a GitHub repository (https://github.com/s-shichino1989/TASSeq-paper). Source data underlying Figs. 4d, h–i and 5a are presented in Supplementary Data 8.

## Code availability

The mapping pipeline for TAS-Seq data is available at https://github.com/s-shichino1989/TASSeq (https://doi.org/10.5281/zenodo.6523558)[57]. All the R code used for this study and rDBEC R package (includes functions of distribution-based error correction and utility functions for this study) are available at https://github.com/s-shichino1989/TASSeq-paper (https://doi.org/10.5281/zenodo.6523560)[58].

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

## Acknowledgements

We thank J. Yasuda for their excellent technical assistance. This work was supported by the Japan Agency for Medical Research and Development PRIME program (S.S., JP21gm6210025) and the Japan Society for the Promotion of Science Grant-in-Aid for Scientific Research on Innovative Areas program (Inflammation Cellular Sociology, 17H06392, K.M.), and Grant-in-Aid for Young Scientists (19K16620, S.S.).

## Author contributions

S.S., S.U., and K.M. designed the study. S.S. performed all the experiments except Lewis lung carcinoma experiment. C.-Y.C. and B.W. performed Lewis lung carcinoma experiment. T.Ok., E.S., K.I, T.S. performed sequencing experiments and contributed to the preprocessing of fastq data. S.S., N.O.S., M.K., and T.I. performed the experiment of human lung samples. S.Ho. performed diagnosis of the existence of fibrosis in RA-ILD samples. N.S. and T.K performed surgical operation of RA-ILD patient. S.S., Y.I., and T.W. performed mouse Kidney experiment. S.S., S.U., S.Ha., T.Og., H.A., W.B., C.C., M.K., T.I., T.Ok., E.S., K.I., T.S., and K.M. wrote the manuscript. K.M. supervised the study.

## Competing interests

S.S. reports advisory role for ImmunoGeneTeqs, Inc; stock for ImmunoGeneTeqs, Inc, S.U. reports advisory role for ImmunoGeneTeqs, Inc; stock for ImmunoGeneTeqs, Inc, IDAC Theranostics, Inc. H.A. reports stock for ImmunoGeneTeqs, Inc., K.M. reports consulting or advisory role for Kyowa-Hakko Kirin, ImmunoGeneTeqs, Inc; research funding from Kyowa-Hakko Kirin, and Ono; stock for ImmunoGeneTeqs, Inc, IDAC Theranostics, Inc., T.I. reports consulting or advisory role for ROHTO Pharmaceutical Co., Ltd; research funding from ROHTO Pharmaceutical Co., Ltd. All the other authors declare no competing interests.
