## [Peer Review file · Communications Biology]

Reviewers' comments:

Reviewer #1 (Remarks to the Author):

Shichino et al proposed an improved TdT based scRNA-seq method, and demonstrated the advantages of the method over others such as Smart-Seq2 and 10x Genomics in capturing more genes, resolving cell composition more accurately, and detecting cell-cell interactions more robustly. There is only limited novelty since the basic methodology, i.e., TdT based scRNA-seq, was proposed previously. However, as the authors claimed, the improved method could facilitate the practical application of TdT based scRNA-seq. Considering that most commonly used scRNA-seq strategies have been suffering the problems of dropout of specific cells and inaccurate representation of composition cells, the study could be important in the field. However, there is a lack of details in the current manuscript, which impedes further evaluation of the applicability of the proposed method.

Major concerns:

1. A lack of direct comparison with other TdT based scRNA-seq methods. The authors only asserted the advantages of TAS-Seq over other TdT based scRNA-seq methods in ABSTRACT and INTRODUCTION, rather than made a comprehensive comparison. The advantages of TdT based scRNA-seq (e.g., Quartz-Seq2) over others, e.g., detection of more genes, were well demonstrated previously (Sasagawa et al., 2018, *Genome Biology*, 19: 29; Sasagawa et al., 2013, *Genome Biology*, 14: R31). Therefore, it not surprising to observe the effect of TAS-Seq in detection of more genes. In contrast, it should be more important to explain the modification and effect improvement of TAS-Seq over Quartz-Seq2 by direct comparison rather than simple description.
2. A lack of comprehensive evaluation of possible biases of TAS-Seq and other methods and their different underlying mechanisms in gene detection. Although TAS-Seq detected more genes with fewer reads than other methods, it did not mean that TAS-Seq detected all genes expressed. As shown in Fig. S6a, genes were also not detected by TAS-Seq. More detailed analysis could be performed to observe whether the undetected genes by TAS-Seq (but detected by other methods) are random or show some biases, and comparison could be further done against the genes undetected by other methods. The biases could be GC content, abundance, expression by specific cells, etc. Together with the larger sensitivity of gene detection, confirmation of the relatively higher randomness of undetected genes could better explain the lower dropout rate of cell types and the consistence of cell type fractions with flow cytometry results. Similarly, TAS-Seq does not always detect more genes than Smart-Seq2 or 10x Genomics. For instance, in Pericytes (Fig. S3), Smart-Seq2 detected significantly more genes than TAS-Seq. Therefore, thorough assessment on the possible biases of TAS-Seq appeared very important.
3. The possible drawbacks of the technique and the current study should be addressed. For example, Quartz-Seq showed the preference for high GC-content genes, and is it same for TAS-Seq? Only lung tissues from human and mouse were assessed in the study, it cannot be guaranteed that the method is also effective in other tissue types.
4. Quantification of the gene expression levels in single cells. With lower sequencing depth, TAS-Seq could capture more genes than Smart-Seq2 and 10x Genomics, also highly variable genes. However, genes in cells are expressed unevenly, with fewest genes expressed most abundantly and most genes expressed lowly or moderately. If the transcripts are captured randomly, abundantly expressed genes should be detected more occasionally, while the large number of genes with low or moderate expression should have much smaller chance to be detected. Therefore, it would be interesting to know the read – gene distribution for Smart-Seq2, 10x Genomics and TAS-Seq. According to the above hypothesis, TAS-Seq could show biases for specific genes. If it were the case, the TAS-Seq quantification of gene expression would not represent the original level in the single cells. A comparison should also be done between TAS-Seq and other methods.

Minor points:

1. "...TAS-Seq could detect more gene numbers and highly-variable genes ..." → "...TAS-Seq could detect more genes and highly-variable genes ...".

Reviewer #2 (Remarks to the Author):

In this study by Shichino et al., the authors devised the method for the cDNA amplification and showed its applicability in scRNA-seq. The authors named this cDNA amplification-based method as TAS-Seq, mainly due to the usage of terminator-assisted solid-phase cDNA amplification and its downstream sequencing. Authors show that TAS-Seq elevates the gene detection sensitivity, and therefore, offers a robust means to explore cell-cell interaction. The authors provided a detailed analysis on two independent lung datasets from murine and humans and observed better outcomes in the case of TAS-Seq. Most importantly, the comparative analysis of TAS-Seq with the SMART-Seq with lower sequencing depth yielded more genes in the case of TAS-Seq, suggesting the higher sensitivity of this method. Overall the method proposed in this manuscript is good and detailed, however, the manuscript requires major formatting. Please check my comments below.

Following are my minor comments:

1. Present format of the manuscript is rather concise, and various sections can be presented with many details.
2. Manuscript lacks a proper discussion section, where the authors might discuss the key limiting steps of TAS-Seq, a constructive discussion about solid-phase cDNA amplification techniques, and their usability in scRNA-seq.
3. I recommend the authors to consider submitting the detailed protocol on structured platforms such as <https://experiments.springernature.com/> or others as this will significantly increase the chances of the adoption of their method by the larger community?
4. In the present format, the main figures are rather scattered and disorganized. I recommend authors reconsider this point and may split the bulky figures into organized multiple figures.
5. Authors have performed their entire analysis using a pretty much older version of the Seurat software suite (Version 2), Any specific reasons? Though many parts of the software have been improved with version 4, therefore, it's worth rechecking their results with the latest version.
6. Authors have depicted the underlying distribution of the number of detected genes in TAS-Seq and other methods and observed more number of genes in the case of TAS-Seq. Notably, unlike other methods, TAS-Seq results indicate a broad distribution. Authors must discuss these findings in the results.

Reviewer #3 (Remarks to the Author):

The manuscript by Shichino et al. describes an improved protocol for generation single-cell mRNA-sequencing libraries and presents comparisons with some of the most used current protocols. The TAS-seq protocol is an evolution of the Quartz-seq protocol, wherein transcripts from captured cell are barcoded early, and amplification of full-length transcripts is ensured using terminal transferase reaction. Final libraries are prepared from the 3'-terminal fragments containing the cellular barcode sequences (although this is not described in detail).

The method promises higher gene detection sensitivity than the most popular current alternatives, addressing a very important limitation of this technology. In addition, the authors demonstrate compatibility with the cell hashing application, further increasing the versatility of the method.

In my opinion, the most important shortcoming of the work is lack of replication and to some extent confounding of biological and technical variation. For example, while the data shown in figure S1 quite

convincingly suggests higher sensitivity for TAS-seq compared with the BD WTA-kit, it is based on a single experiment. More importantly, the comparisons between different methods are based on datasets produced in different labs at different times. Therefore, while they are based on same tissue and similar processing protocols, it is difficult to assess the effect of manual batch variation from biases inherently related to the different technologies. For example, in Figure 2g markedly different cell type frequencies are observed with the v2 and v3 generations of the 10x Genomics technology. I find this surprising as both kits are based on the same instrument and similar microfluidic cell partitioning process. Furthermore, as the comparison is based on single data sets, there are no error bars or estimates for statistical significance in this case.

The ideal way to carry out such comparisons would of course involve performing all analyses at the same time using the same sample material, which of course is very challenging to execute in practice. But nevertheless, I would respectfully encourage the authors to consider either finding more public datasets that could be included in the comparisons, producing additional replicate data using TAS-seq, or toning down the conclusions and discussing the limitations of this work in more detail.

Minor comments:

Page 2: "human neutrophils dropout frequently occurs in 10X Chromium system"

This should be discussed in more detail, explaining why the 10X Chromium might be particularly susceptible for losing these sensitive cells.

Page 4: "We found that TAS-Seq detected more genes (3026 genes/50000 reads) than BD WTA kit (1997 genes/50000 reads)"

Are these average or perhaps median values? In figure S1b, it would be useful to have an additional violin plot or a histogram showing the read depth distribution in both samples.

Figure 2b: why is the distribution so different between TAS-seq and 10x genomics? (and to lesser extent smart-seq2)

Figure 2c: why the 10x technology is not included in this plot? In my opinion it would be arguably a more interesting comparison, as both methods are based on 3'-counting (whereas smart-seq2 libraries contain fragments along the entire length of a transcript, warranting deeper sequencing)

Figures 2f and S4: some text is not readable

Figure S7: I personally find the cumulative gene expression counts somewhat difficult to interpret and would suggest the authors to consider e.g. a dot plot that would visualize both gene expression frequency and gene expression intensity for each cell type. The fact that Csf2 gene is readily detected in ILC2 using both technologies suggests to me that the reason for these observed differences might also be related to the biological properties of the samples, or subtle differences in cell clustering. I hope the authors can shed more light on this.

Methods section does not contain all details on the sequencing configuration, most importantly read lengths

There are some typos and some parts of the manuscript would benefit from language check.

Point-by-point responses to Reviewer's comments in revised manuscript

Response to Reviewer #1

We deeply appreciate the insightful comments from Reviewer #1, which helped us to improve the quality of our study and our manuscript.

Major concerns:

1. A lack of direct comparison with other TdT based scRNA-seq methods. The authors only asserted the advantages of TAS-Seq over other TdT based scRNA-seq methods in ABSTRACT and INTRODUCTION, rather than made a comprehensive comparison. The advantages of TdT based scRNA-seq (e.g., Quartz-Seq2) over others, e.g., detection of more genes, were well demonstrated previously (Sasagawa et al., 2018, Genome Biology, 19: 29; Sasagawa et al., 2013, Genome Biology, 14: R31). Therefore, it not surprising to observe the effect of TAS-Seq in detection of more genes. In contrast, it should be more important to explain the modification and effect improvement of TAS-Seq over Quartz-Seq2 by direct comparison rather than simple description.

Response:

Thank you for pointing out the necessity for more detailed explanation of the differences between TAS-Seq and Quartz-seq2. To solve the issue, we evaluated the tolerance of Quartz-seq2 against amount of TdT enzyme and reaction time in terms of excess elongation of primer-derived by-products. We found that the length of remaining primer-derived products was less than 200 bp under standard Quartz-seq2 conditions, similar to that of TAS-Seq (Fig. 2b). When using 1.5 times of the amount of TdT, the amount of primer-derived by-products increased, but the size distribution did not change compared to that yielded under standard conditions (Fig. 2b). We also found that, when the TdT reaction was pre-incubated at 23°C for 5, 10, and 15 minutes (simulating the case of accidentally warming the TdT reaction to room temperature), the length of primer-derived by-products extended over 200 bp, depending on incubation time (Fig. 2b). This suggests that our TAS-Seq approach is more tolerable of variable reaction conditions than Quartz-seq2 in terms of suppressing excessive extension and production of primer-derived products. In comparison, TAS-Seq did not alter the extension of primer-derived by-products within the reaction time range from 5 min to 45 min, nor under a 4.2-fold increase in TdT enzyme amount (Fig. 2a). These results indicated that TAS-Seq exhibits enhanced the tolerance and reduced handling difficulty of the TdT reaction compared to that of Quartz-seq2. We added the above results and associated explanation to our revised manuscript in the results section (page 5, line 24 to line 34) and newly-added discussion section (page 11, line 10 to line 17).

2. A lack of comprehensive evaluation of possible biases of TAS-Seq and other methods and their different underlying mechanisms in gene detection. Although TAS-Seq detected more genes with fewer

reads than other methods, it did not mean that TAS-Seq detected all genes expressed. As shown in Fig. S6a, genes were also not detected by TAS-Seq. More detailed analysis could be performed to observe whether the undetected genes by TAS-Seq (but detected by other methods) are random or show some biases, and comparison could be further done against the genes undetected by other methods. The biases could be GC content, abundance, expression by specific cells, etc. Together with the larger sensitivity of gene detection, confirmation of the relatively higher randomness of undetected genes could better explain the lower dropout rate of cell types and the consistence of cell type fractions with flow cytometry results. Similarly, TAS-Seq does not always detect more genes than Smart-Seq2 or 10x Genomics. For instance, in Pericytes (Fig. S3), Smart-Seq2 detected significantly more genes than TAS-Seq. Therefore, thorough assessment on the possible biases of TAS-Seq appeared very important.

Response:

Thank you for noting the necessity of comprehensive analysis of possible gene-detection bias in TAS-Seq compared with that in the other methods. We analyzed the distribution of positive cells (for evaluation of the drop-out rate of each gene), average expression level, percent GC content, and transcript length of each gene in TAS-Seq, 10X v2, and Smart-seq2 datasets from murine lung cell subsets. We found that drop-out rates of commonly-detected genes between TAS-Seq and 10X v2 or Smart-seq2 were lower in TAS-Seq than in the other platforms (Supplementary Fig. 6a and 6b). In addition, genes detected only in TAS-Seq exhibited lower expression than all other genes in each murine lung cell subset (Supplementary Fig. 6a and 6b), suggesting that TAS-Seq is more sensitive to low gene expression than 10X v2 and Smart-seq2 in cell subsets of steady-state adult murine lung tissue. We analyzed the distribution of percent GC content and transcript length for the genes detected more favorably by TAS-Seq than by 10X v2 or Smart-seq2. We found that the variance of GC content distribution was similar between TAS-Seq favorable genes and commonly-detected genes (Supplementary Fig. 8), whereas the transcript length distribution tended to be greater in TAS-Seq favorable genes than in commonly-detected genes (Supplementary Fig 9). This suggests that TAS-Seq maintains randomness of gene GC content and increases randomness of gene transcript length, even in poorly-expressed genes in murine lung cell subsets. This quality may contribute to the accuracy of the cell composition analysis of the murine lung TAS-Seq data. Notably, TAS-Seq–favored genes included more AT-rich genes (Supplementary Fig. 8) and genes with long transcripts (Supplementary Fig. 9) than commonly-detected genes, suggesting that AT-rich and long–transcript genes are more favorably detected by TAS-Seq than by 10X v2 and Smart-seq2 in murine lung cell subsets. We added these results and the associated explanations to the results (page 8, line 21 to page 9, line 4) and discussion sections (page 12, line 15 to 34) of the revised manuscript.

In addition, when read depth of new TAS-Seq datasets was adjusted to 1/5 that of the Smart-seq2 dataset, the number of detected genes by TAS-Seq was higher than that detected by Smart-seq2 in

murine lung pericytes (Supplementary Fig.6). In the old manuscript, the read depth of TAS-Seq datasets was 1/10 that of the Smart-seq2 dataset, and was not sufficient to detect a high number of genes using TAS-Seq in murine lung pericytes. Compared to the other cell subsets, the difference of the detected gene number between TAS-Seq and Smart-seq2 was small for murine lung pericytes (Supplementary Fig. 6). This may be due to differences in the nature of expressed genes among lung cell subsets, possibly including transcript length and percent GC content. However, that point is beyond the scope of this study, and we did not include this discussion in the revised manuscript.

3. The possible drawbacks of the technique and the current study should be addressed. For example, Quartz-Seq showed the preference for high GC-content genes, and is it same for TAS-Seq?

Response:

Thank you for advising us to describe the possible drawbacks of TAS-Seq. First, in contrast to Quartz-seq, which preferentially detects GC-rich genes, we found that TAS-Seq favored AT-rich genes (Supplementary Fig.8), as mentioned in the above response to your Major comment #2. One of the major differences between TAS-Seq and Quartz-seq is that TAS-Seq uses polyC-tailing, but Quartz-seq uses polyA-tailing for the TdT reaction. In the TdT reaction process, short polyN tailed cDNA given a hairpin structure by intra- or inter-molecular annealing of the polyN tail to the intermediate transcript region could be less efficiently extended by TdT because TdT-tailing efficiently is dropped down against the recessive end of double-stranded DNA (Deng et al Nucleic Acid Res 1981). The tagging primer for second strand synthesis may less efficiently anneal and extend against such cDNAs than cDNAs with long polyN tail without intra- or inter-molecular annealing. Hence, AT- and GC-rich genes may be hard to amplify by polyA- and polyC-based TdT cDNA amplification, explaining the detection bias of AT- and GC-rich genes in TAS-Seq and Quartz-seq. We added this explanation to discussion section (page 12 line 22 to line 34). Second, the TAS-Seq approach theoretically limits tagging efficiency of the second strand synthesis reaction more so than existing TdT-based scRNA-seq methods because short polyN-tailed cDNAs, which might not have sufficient polyN tail to anneal tagging primers, are stochastically produced by dideoxynucleotide spike-in. We also added this possible drawback to the discussion section (page 11, line 23 to line 27).

Only lung tissues from human and mouse were assessed in the study, it cannot be guaranteed that the method is also effective in other tissue types.

Response:

Thank you for pointing out this concern regarding the utility of TAS-Seq. We compared data for adult murine spleen and kidney samples, and found that TAS-Seq outperformed 10X v2 and Smart-seq2 in

these organs in terms of gene detection sensitivity and detection of highly-variable genes (Supplementary Fig. 3). However, because of limited experimental resources, we could not test all of the organs included in the Tabula Muris consortium. Therefore, we also added an explanation of the study limitations in terms of the organ-wide/context-wide utility of TAS-Seq in the discussion section to recommend further evaluation of the utility of TAS-Seq (page 13, line 12 to line 15).

4. Quantification of the gene expression levels in single cells. With lower sequencing depth, TAS-Seq could capture more genes than Smart-Seq2 and 10x Genomics, also highly variable genes. However, genes in cells are expressed unevenly, with fewest genes expressed most abundantly and most genes expressed lowly or moderately. If the transcripts are captured randomly, abundantly expressed genes should be detected more occasionally, while the large number of genes with low or moderate expression should have much smaller chance to be detected. Therefore, it would be interesting to know the read – gene distribution for Smart-Seq2, 10x Genomics and TAS-Seq. According to the above hypothesis, TAS-Seq could show biases for specific genes. If it were the case, the TAS-Seq quantification of gene expression would not represent the original level in the single cells. A comparison should also be done between TAS-Seq and other methods.

Response:

Thank you for requesting evaluation of the evenness of gene detection by expression level via TAS-Seq. We analyzed the distribution of read counts of detected genes in pseudo-bulk data of each murine lung cell subset from TAS-Seq, 10X v2, and Smart-seq2 datasets. We found that major parts of detected genes were highly expressed in 10X v2 and Smart-seq2 data (Supplementary Fig. 10a). In contrast, the read count of detected genes of TAS-Seq data distributed more uniformly against gene-expression levels, and poorly-expressed genes were detected more frequently in deep-sequenced TAS-Seq data (Supplementary Fig, 10a). We further compared the distribution of lung fibroblasts with that of bulk RNA-seq data from sorted murine lung fibroblasts using our previously-published data (GSE110540). Similarity calculation of the density distribution via Kullback-Leibler divergence revealed that the similarity between TAS-Seq and bulk RNA-seq data was higher than that between 10X v2 or Smart-seq2 and bulk RNA-seq data (Supplementary Fig. 10b and 10c). These results indicated that TAS-Seq could detect genes across a wide range of expression more uniformly than 10X v2 or Smart-seq2. We added these results to the results (page 8, line 19 to page 9, line 11) and discussion sections (page 11, line 18 to line 27).

Minor points:

1. “...TAS-Seq could detect more gene numbers and highly-variable genes ...” → “...TAS-Seq could detect more genes and highly-variable genes ...”.

Response:

Thank you for pointing out the erroneous description. We corrected the sentence accordingly.

Response to Reviewer #2

We deeply appreciate the insightful comments from Reviewer #2, which helped us improve the quality of our study and our manuscript.

Following are my minor comments:

1. Present format of the manuscript is rather concise, and various sections can be presented with many details.

Response:

We apologize the previous manuscript format was rather concise; we initially intended to submit a short communication. To improve readability and informativity of our manuscript, we re-formatted in the standard manuscript style, including a sub-sectioned results section with more detailed explanation of each result.

2. Manuscript lacks a proper discussion section, where the authors might discuss the key limiting steps of TAS-Seq, a constructive discussion about solid-phase cDNA amplification techniques, and their usability in scRNA-seq.

Response:

We apologize for this missing information. We added a discussion section mentioning the key limitations of TAS-Seq (page 13, line 12-22), along with a constructive discussion about solid-phase cDNA amplification techniques and their applicability to scRNA-seq. (page 13 line 23 to line 33). We also included information about other TdT-based scRNA-seq methods in the introduction section (page 3, line 22 to line 33).

3. I recommend the authors to consider submitting the detailed protocol on structured platforms such as <https://experiments.springernature.com/> or others as this will significantly increase the chances of the adoption of their method by the larger community?

Response:

Thank you for your suggestion to enhance the accessibility of our TAS-Seq method. We are planning to post a step-by-step protocol on structured platforms, including any nature portfolio or open-source protocols.io database, to improve access to our data. We have reached out to the Communication Biology editors regarding this.

4. In the present format, the main figures are rather scattered and disorganized. I recommend authors

reconsider this point and may split the bulky figures into organized multiple figures.

Response:

We apologize for the scattered and disorganized figures. We rearranged our result figure contents and divided them into more meaningful blocks to make each figure easier to understand. We also created results subheadings associated with these changes.

5. Authors have performed their entire analysis using a pretty much older version of the Seurat software suite (Version 2), Any specific reasons? Though many parts of the software have been improved with version 4, therefore, it's worth rechecking their results with the latest version.

Response:

Thank you for pointing out the necessity to check our results using the newer version of Seurat software. We re-formatted our analysis pipeline to Seurat v4 and re-analyzed all data in the manuscript. We found that the cell clustering results were similar to those yielded by Seurat v2 analysis. We replaced the results with Seurat v4-generated results in the revised manuscript.

6. Authors have depicted the underlying distribution of the number of detected genes in TAS-Seq and other methods and observed more number of genes in the case of TAS-Seq. Notably, unlike other methods, TAS-Seq results indicate a broad distribution. Authors must discuss these findings in the results.

Thank you for pointing out the necessity to comprehensively analyze the possible gene-detection distribution of TAS-Seq data compared with that yielded by other methods. We first analyzed the distribution of read counts of detected genes in pseudo-bulk data for each murine lung cell subset from TAS-Seq, 10X v2, and Smart-seq2 analysis. We found that major parts of detected genes were highly expressed in 10X v2 and Smart-seq2 data (Supplementary Fig. 10a). In contrast, the read count of detected genes in TAS-Seq data were distributed more uniformly against gene-expression levels than that for 10X v2 and Smart-seq2 among analyzed cell subsets, and weakly-expressed genes were detected more frequently in deep-sequenced TAS-Seq data (Supplementary Fig. 10a). We further compared the distribution of lung fibroblasts with that of bulk RNA-seq data from sorted murine lung fibroblasts using our previously-published data (GSE110540) Similarity calculation of the density distribution via Kullback-Leibler divergence revealed that the similarity between TAS-Seq and bulk RNA-seq data was higher than that between 10X v2 or Smart-seq2 and bulk RNA-seq data (Supplementary Fig. 10b and 10c). These results indicated that TAS-Seq detected expressed genes across a range of expression more uniformly than 10X v2 and Smart-seq2. We added these results to

the results (page 8, line 35 to page 9, line 11) and discussion sections (page 11, line 21 to line 23).

We also analyzed the distribution of the percent of positive cells (to evaluate the drop-out rate of each gene), average expression level, percent GC content, and transcript length of each gene in the TAS-Seq, 10X v2, and Smart-seq2 datasets of murine lung cell subsets. We found that drop-out rates of commonly-detected genes between TAS-Seq and 10X v2 or Smart-seq2 were lower in TAS-Seq than the other platforms (Supplementary Fig. 6a and 6b). In addition, genes detected only by TAS-Seq were more weakly expressed than those in other murine lung cell subsets (Supplementary Fig. 6a and 6b), suggesting that TAS-Seq is more sensitive to low gene expression than 10X v2 and Smart-seq2. We analyzed the distribution of percent GC content and transcript length of the genes detected more favorably by TAS-Seq than by 10X v2 or Smart-seq2. We found that the variance of GC content distribution was similar between TAS-Seq favorable genes and commonly-detected genes (Supplementary Fig. 8), and that transcript length distribution was greater in TAS-Seq favorable genes than in commonly-detected genes (Supplementary Fig 9), suggesting that TAS-Seq maintains randomness of gene GC content while increasing randomness of gene transcript length, even in the weakly-expressed genes of murine lung cell subsets. This may contribute to the accuracy of cell composition analysis of the TAS-Seq data. Notably, TAS-Seq favorable genes included significantly more AT-rich genes (Supplementary Fig. 8) and more long transcripts (Supplementary Fig. 9) than commonly-detected genes, suggesting that AT-rich and long-transcript genes are more favorably detected by TAS-Seq than by 10X v2 and Smart-seq2 in murine lung cell subsets. We added these results and associated explanations to the results (page 8, line 19 to line 34) and discussion sections (page 12, line 15 to line 34) of the revised manuscript.

Response to Reviewer #3

We deeply appreciate the insightful comments from Reviewer #3 that helped us to significantly improve the quality of our study and our manuscript.

Final libraries are prepared from the 3'-terminal fragments containing the cellular barcode sequences (although this is not described in detail).

Response:

Thank you for pointing out the lack of the explanation of final library construction process in the TAS-Seq scheme image. We added the final library structure figure to Figure 1 in the revised manuscript.

In my opinion, the most important shortcoming of the work is lack of replication and to some extent confounding of biological and technical variation. For example, while the data shown in figure S1 quite convincingly suggests higher sensitivity for TAS-seq compared with the BD WTA-kit, it is based on a single experiment. More importantly, the comparisons between different methods are based on datasets produced in different labs at different times. Therefore, while they are based on same tissue and similar processing protocols, it is difficult to assess the effect of manual batch variation from biases inherently related to the different technologies. For example, in Figure 2g markedly different cell type frequencies are observed with the v2 and v3 generations of the 10x Genomics technology. I find this surprising as both kits are based on the same instrument and similar microfluidic cell partitioning process.

Furthermore, as the comparison is based on single data sets, there are no error bars or estimates for statistical significance in this case.

The ideal way to carry out such comparisons would of course involve performing all analyses at the same time using the same sample material, which of course is very challenging to execute in practice. But nevertheless, I would respectfully encourage the authors to consider either finding more public datasets that could be included in the comparisons, producing additional replicate data using TAS-seq, or toning down the conclusions and discussing the limitations of this work in more detail.

Response:

Thank you for point out the major limitation of the previous manuscript. To partly address these issues, we performed a technical replicate experiment with TAS-Seq data of mouse lung and spleen to evaluate technical variations. We also changed the data analysis scheme for 10X v2 data of the Tabula Muris Consortium because the 10X data included four independently-collected datasets (two of whole lung cell data, others of stromal cell-enriched data) taken from four different mice, and we mistakenly pooled the different datasets. We used 10X v2 datasets of the whole lung cell data from Tabula Muris,

and one additional 10X v2 dataset of whole lung cell data (GSM3926540). We also checked the quality of 10X v3 data (GSE145998) used in the previous version of the manuscript to seek a possible source of the discrepancy between 10X v2 and 10X v3 datasets. We found that the GSE145998 data contained too many cells in one 10X v3 lane (39558 cells were detected by CellRanger-6.1.2), possibly distorting scRNA-seq data with doublets and RNA leaks.

We searched other publicly-available datasets and checked sample characteristics and library quality. Unfortunately, we could not find 10X v2 or v3 scRNA-seq data of non-enriched whole murine lung cells with sufficient library quality (sufficient read number or cells were not overloaded). Thus, we performed comparison analysis using three 10X v2 datasets, one Smart-seq2 dataset, and three TAS-Seq datasets from mouse lung tissue in the revised manuscript. We found that TAS-Seq detected more overall and highly-variable genes in mouse lung tissue with minimal technical variability (Figure 3). TAS-Seq data also showed high correlation with flow-cytometric data (Figure 4) and included a high number of inferred cell-cell communications (Figure 5). We also compared mouse spleen and kidney datasets, and acquired similar results to those for mouse lungs (Supplementary Figure 3). We added the above results to the results section in the revised manuscript (page 6, line 22 to page 7, line 19).

We completely agree about the limitation of the comparison of different platforms using public datasets because the biological material and cell preparation techniques are not consistent between our data and publicly-available data. We should use exactly the same materials/equipment simultaneously to perform an accurate comparison of different platforms. According to your suggestion, we mentioned the limitations of our study in discussion section (page 13, line 12-22), and mentioned uncertainty in related result statements.

Minor comments:

Page 2: “human neutrophils dropout frequently occurs in 10X Chromium system”

This should be discussed in more detail, explaining why the 10X Chromium might be particularly susceptible for losing these sensitive cells.

Response:

Thank you for your suggestion. One of the possible mechanisms by which 10X Chromium dropped out human neutrophils is that the cell suspension was mixed with nuclease-free water, which changes osmotic pressure and possibly damages neutrophils in a 10X workflow. The other possibility is that RNase derived from neutrophils led to degradation of RNases in the droplet of the 10X system. The BD Rhapsody system, but not the 10X system, can use a strong, detergent-based lysis buffer that efficiently inactivate RNases because the lysis buffer is completely washed out before reverse transcription. We added this information to the discussion section of the revised manuscript (page 11, line 28 to page 12, line 14).

Page 4: “We found that TAS-Seq detected more genes (3026 genes/50000 reads) than BD WTA kit (1997 genes/50000 reads)”

Are these average or perhaps median values? In figure S1b, it would be useful to have an additional violin plot or a histogram showing the read depth distribution in both samples.

Response:

Thank you for noting the missing information needed to interpret our results. We explained detected gene number and read depth information more clearly regarding newly collected spleen datasets, which contained three technical replicates per group, in the revised manuscript as follows: “We found that TAS-Seq significantly detected more genes (median 1929.3 ± 24.1 genes at median 15486.6 ± 18.6 reads/cell) than BD WTA kit (median 1595.6 ± 76.0 genes at median 15242.6 ± 77.6 reads/cell) (Supplementary Fig. 2b).” (page 6, line 24-26).

Figure 2b: why is the distribution so different between TAS-seq and 10x genomics? (and to lesser extent smart-seq2)

Response:

Thank you for this important question regarding the difference in murine lung cell composition obtained by TAS-Seq, 10X v2, and Smart-seq2. We could not identify the significant sources that led to this discrepancy. Instead, we added some possible reasons to the discussion section of the revised manuscript as follows: “These characteristics possibly come from the difference of the cell capture strategies; i.e. TAS-Seq used BD Rhapsody nanowell system and cells were collected by gravity flow in isotonic buffer, that might more mild isolation condition for cells than 10X system (cell suspension is mixed with nuclease-free water that changes osmotic pressure, which possibly damages cells) and Smart-seq2 (cells are isolated into microtiter plates by cell sorter that induces mechanical damage to cells). The other possibility that might lead such difference is that BD Rhapsody system, but not 10X system, could use strong detergent-based lysis buffer that efficiently inactivate RNases because the lysis buffer could be completely washed out before reverse transcription reaction. These characteristics of nanowell system possibly become TAS-Seq more suitable for analyzing fragile and RNase-rich cells than droplet or cell sorter-based scRNA-seq platforms. Another possibility of the source of the accuracy of cell composition of TAS-Seq data was broader detection of lowly-expressed genes and longer genes than 10X v2 and Smart-seq2, which could lead efficient detection cells which have small amount of mRNA.” (page 11, line 28 to page 12, line 14)

Figure 2c: why the 10x technology is not included in this plot? In my opinion it would be arguably a

more interesting comparison, as both methods are based on 3'-counting (whereas smart-seq2 libraries contain fragments along the entire length of a transcript, warranting deeper sequencing)

Response:

Thank you for pointing out this important issue. In the previous version of the manuscript, we used UMI count data created by the Tabula Muris consortium, so the actual raw count information for each cell was missing. To address this issue, we retrieved raw fastq files of 10X v2 Tabula Muris data from processed BAM files, then re-mapped using CellRanger-6.1.2. Next, raw count (non-UMI) gene-expression matrixes were created from mapped BAM data and filtered by valid cell barcodes identified using CellRanger-6.1.2. Based on the raw-count matrix data, we added 10X v2 data to the scatter plot of gene/read distribution in the revised manuscript (Figure 3b, 3d and Supplementary Figure 3a and 3c).

Figures 2f and S4: some text is not readable

Response:

We apologize for the improper text labels. We corrected the text of cell cluster labels to improve readability.

Figure S7: I personally find the cumulative gene expression counts somewhat difficult to interpret and would suggest the authors to consider e.g. a dot plot that would visualize both gene expression frequency and gene expression intensity for each cell type. The fact that Csf2 gene is readily detected in ILC2 using both technologies suggests to me that the reason for these observed differences might also be related to the biological properties of the samples, or subtle differences in cell clustering. I hope the authors can shed more light on this.

Response:

Thank you for insightful suggestion. We agree that dotplot visualization of the patterns of expression intensity and positivity for each gene in each lung cell subset is easier to interpret. Based on your suggestion, we searched for biological differences in the samples between our TAS-Seq data and 10X v3 data (GSE145998), which was analyzed in previous manuscript. In addition to the cell overload problem of the murine lung 10X v3 data analyzed in previous manuscript, we found that GSE145998 data used Collagenase D for the enzymatic dissociation of the lung samples, which contains a significant amount of endotoxin that could change the expression pattern of inflammatory-related genes. However, we and the Tabula Muris consortium used the low-endotoxin dissociation enzyme Liberase TM to avoid such artifacts. Therefore, we removed the 10X v3 data from revised manuscript

and compared our TAS-Seq data with the Tabula Muris 10X v2 and Smart-seq2 datasets using dotplot visualization. As in the previous version of our manuscript, we found that TAS-Seq more broadly detected a wider range of growth factors among lung cell subsets, including BMPs, Csf1, FGFs, PDGFs, TGF, and VEGFs, than the other datasets, and Pdgfb, Fgf7, and Fgf10 exhibited higher positivity in the TAS-Seq data than in the 10X v2/Smart-seq2 datasets (Fig. 6a). We also found that TAS-Seq and 10X v2 data, but not Smart-seq2 data, showed broad expression of IL-2, an important cytokine for T cell activation, in lung cell subsets (Fig. 6b). In addition, only TAS-Seq revealed broad expression of IL-11, an important cytokine in lung injury and fibrosis, in various lung cell subsets, consistent with a previous report (Traber et al. PLoS One 2019). These data suggest that TAS-Seq captures expression patterns of growth factors/interleukins and their significant cellular sources more robustly than 10X v2 and Smart-seq2 in steady-state murine lungs. We added the above results to the revised version of the manuscript (page 10, line 7-27).

Methods section does not contain all details on the sequencing configuration, most importantly read lengths

Response:

We apologize for missing important information on the sequencing experiment. We added the sequencing configuration information to the methods section (page 21, line 33-36).

There are some typos and some parts of the manuscript would benefit from language check.

Response:

We apologize the typographical and grammatical errors. We sent our revised manuscript to the language editing service Editage.

REVIEWERS' COMMENTS:

Reviewer #1 (Remarks to the Author):

The authors answered my concerns satisfactorily.

Reviewer #2 (Remarks to the Author):

In the revised manuscript, the authors have addressed all my concerns.

Reviewer #3 (Remarks to the Author):

I thank the authors for diligently addressing most of the concerns that I have previously raised. Overall the manuscript is now much improved and the conclusions are in agreement with the presented data. I would have still wished for more biological and technical replicates, but I do understand that these are difficult and expensive to produce in any bigger scale. In my opinion, the manuscript can be considered for publications with the following remarks:

1. Page 7 lines 31-32

The authors state that TAS-seq has high reproducibility, based on the three technical replicates presented.

In my opinion, this comparison is biased, because the TAS-seq datasets are not truly independent, since the first steps of the protocol were done before the sample was aliquoted into replicates (as described Figure 3A). Therefore it is not surprising that the cell type composition is more similar than in the 10x Genomics datasets, which were produced completely separately. An analogous experiment would have been one where the 10x genomics emulsions were aliquoted post harvesting and the library prep then carried out independently for the aliquots. It might well be that TAS-seq has high reproducibility in terms of cell type coverage, but in my opinion this data alone is not enough to prove it.

2. There are still some typos, such as:

Page 5 line 33: "Quanrt-seq2"

Page 7 line 23: "identified byIn"

Figure 1: "whole-transcirptome"

Response to Reviewer #3

We deeply appreciate the insightful additional comments from Reviewer #3, which helped us to improve the quality of our study and our manuscript.

1. Page 7 lines 31-32

The authors state that TAS-seq has high reproducibility, based on the three technical replicates presented.

In my opinion, this comparison is biased, because the TAS-seq datasets are not truly independent, since the first steps of the protocol were done before the sample was aliquoted into replicates (as described Figure 3A). Therefore it is not surprising that the cell type composition is more similar than in the 10x Genomics datasets, which were produced completely separately. An analogous experiment would have been one where the 10x genomics emulsions were aliquoted post harvesting and the library prep then carried out independently for the aliquots. It might well be that TAS-seq has high reproducibility in terms of cell type coverage, but in my opinion this data alone is not enough to prove it.

We thank you for pointing out an important issue. We agree that more biological/technical replicates of data are necessary to evaluate the reproducibility of TAS-Seq and other technologies. Unfortunately, we could not perform such an extensive analysis because of the limitation of research budget. Thus, according to your suggestion, we removed the phrase

“suggesting that TAS-Seq has high reproducibility” in the revised manuscript (Page 7, line 34). In addition, we have supplemented an additional biological replicate of TAS-Seq data of murine lung, which was presented in the first version of our manuscript, for the comparison of cell composition of murine lungs (Figure 4a-4d and Supplementary Figure 5). We found that the difference in cell composition between the biological replicates of data (TAS-Seq.dataset2 and the other TAS-Seq datasets) was higher than that among the technical replicates of data, but the tendency of cell composition was still similar (Fig. 4b and 4d). In addition, the Pearson’s correlation coefficient was still high for biological replicate data ($R^2 = 0.912$), and the slope of the regression lines of TAS-Seq data was nearest to 1 among the compared platforms. We have added these results in the revised manuscript (Page 7, line 34 to Page 8, line 7).

2. There are still some typos, such as:

Page 5 line 33: “Quanrt-seq2”

Page 7 line 23: “identified byln”

Figure 1: “whole-transcriptome”

Thank you for pointing out typos. We corrected these typos and missing sentences.